# Lectin binding and gel secretion within Lorenzinian electroreceptors of *Polyodon*

**David F. Russell**[1]*, **Wenjuan Zhang**[2], **Thomas C. Warnock**[3], **Lilia L. Neiman**[4]

**1** Department of Biological Sciences, Neuroscience Program, Department of Physics and Astronomy, Ohio University, Athens, Ohio, United States of America, **2** Honors Tutorial College, Ohio University, Athens, Ohio, United States of America, **3** Department of Physics and Astronomy, Ohio University, Athens, Ohio, United States of America, **4** Department of Biological Sciences, Ohio University, Athens, Ohio, United States of America

* russeld2@gmail.com, russeld2@ohio.edu

**Data Availability Statement:** All relevant data are within the paper and its Supporting information files.

## Abstract

We imaged the carbohydrate-selective spatial binding of 8 lectins in the ampullary organs (AOs) of electroreceptors on the rostrum of freshwater paddlefish (*Polyodon spathula*), by fluorescence imaging and morphometry of frozen sections. A focus was candidate sites of secretion of the glycoprotein gel filling the lumen of AOs. The rostrum of *Polyodon* is an electrosensory appendage anterior of the head, covered with >50,000 AOs, each homologous with the ampulla of Lorenzini electroreceptors of marine rays and sharks. A large electrosensory neuroepithelium (EN) lines the basal pole of each AO's lumen in *Polyodon*; support cells occupy most (97%) of an EN's apical area, along with electrosensitive receptor cells. (1) Lectins WGA or SBA labeled the AO gel. High concentrations of the N-acetyl-aminocarbohydrate ligands of these lectins were reported in canal gel of ampullae of Lorenzini, supporting homology of *Polyodon* AOs. In cross sections of EN, WGA or SBA labeled cytoplasmic vesicles and organelles in support cells, especially apically, apparently secretory. Abundant phalloidin + microvilli on the apical faces of support cells yielded the brightest label by lectins WGA or SBA. In parallel views of the apical EN surface, WGA labeled only support cells. We concluded that EN support cells massively secrete gel from their apical microvilli (and surface?), containing amino carbohydrate ligands of WGA or SBA, into the AO lumen. (2) Lectins RCA120 or ConA also labeled EN support cells, each differently. RCA120-fluorescein brightly labeled extensive Golgi tubules in the apical halves of EN cells. ConA did not label microvilli, but brightly labeled small vesicles throughout support cells, apparently non-secretory. (3) We demonstrated "sockets" surrounding the basolateral exteriors of EN receptor cells, as candidate glycocalyces. (4) We explored whether additional secretions may arise from non-EN epithelial cells of the interior ampulla wall. (5) Model: Gel is secreted mainly by support cells in the large EN covering each AO's basal pole. Secreted gel is pushed toward the pore, and out. We modeled gel velocity as increasing ~11x, going distally in AOs (toward the narrowed neck and pore), due to geometrical taper of the ampulla wall. Gel renewal and accelerated expulsion may defend against invasion of the AO lumen by microbes or small parasites. (6) We surveyed lectin labeling of accessory structures, including papilla cells in AO necks, striated ectoderm epidermis, and sheaths on afferent axons or on terminal glia.

**Funding:** Supported by NIH grant 5R21GM103494 to DFR, by research funds from Ohio University, and by a Provost Undergraduate Research Fund grant to WZ. The funders had no role in study design, data collection and analysis, decision to publish, or preparation of the manuscript.

**Competing interests:** The authors have declared that no competing interests exist.

**Abbreviations:** ~, Approximate; Ab, Antibody; ALLn, Anterior Lateral Line nerve; bilateral cranial nerves containing sensory neuron axons (myelinated afferents) to electroreceptors on the rostrum, head, jaws, and gill covers; AO, Ampullary organ; AoL, Ampulla of Lorenzini; -b, Biotinylated lectin; DAPI, 4′,6-diamidino-2-phenylindole; blue-fluorescent intercalation dye for nuclear DNA; EM, Electron microscopy; EN, Electrosensory neuroepithelium; ER, Electroreceptor; ID (OD), Inside (outside) diameter; k, Kinocilium, one per receptor cell; NA, Numerical aperture; PBS, Phosphate buffered saline; R, Receptor cell; S, Support cell; SD, Standard deviation (v–1); t, Number of probe applications; v, Number of measurements; zo, Zona occludens. Table 1 lists abbreviations of lectin names; μ, Micron, micrometer, 1 x 10$^{-6}$ meter.

## Introduction

Cutaneous secretion is of general interest in biomedical sciences, e.g., secretion of sebum along hair follicles. Ducted cutaneous secretion occurs prominently in ampullary electroreceptors (ERs). These specialized sensory receptors in the skin of certain aquatic vertebrate animals sense microvolt-scale external electrical signals, e.g., from prey [1]. Such ERs have one or more ampullary organs (AO): flask-shaped invaginations of skin, whose deepest part is lined by an electrosensory neuroepithelium (EN) containing an array of electrosensitive receptor cells. A canal connects an AO's lumen to a skin pore; the canal is generally short in freshwater species or long in marine species.

The hollow lumen of the ampulla and long canal is filled with an electrically conductive gel, secreted internally. Canal gel has been most studied in the large ampulla of Lorenzini (AoL) of marine elasmobranch fishes [1, 2], an ancestral type of electroreceptor. Chemical constituents of AoL gel include N-acetyl amino carbohydrates, sulfated polysaccharides including keratan sulfates, and mucus and other proteins [3–7]. The composition of AoL gel varies between elasmobranch species [5].

We applied lectins (Table 1) to fluorescently label the ampullary electroreceptors of freshwater paddlefish (*Polyodon spathula*). Lectins are globular proteins, most from plants, with recognition domains for carbohydrates attached to macromolecules or cell surfaces. Since *Polyodon*'s ampullary organs are homologous with elasmobranch AoL based on comparative, genomic, and fate-mapping data [8–11], and since the gel in AoL canals contains certain N-acetyl amino carbohydrates at high concentrations [4–7], corresponding lectins were natural choices for labeling *Polyodon* AOs.

Camacho et al. [12] reported EM and carbohydrate staining of AOs and gel in a sturgeon (related to *Polyodon* in the family Acipenseriformes). They concluded that AO gel is secreted by support cells of the electrosensory neuroepithelium. In marine AoL, EM imaging indicated gel secretion from the internal wall epithelia of canals, and in the ampulla also [2, 13].

*Polyodon* has a rostrum, a flattened electrosensory appendage, anterior of the head. The rostrum and head are covered with specialized skin in which >50,000 AOs are embedded [14, 15]. Clusters of ~23 adjacent AOs form the receptive field of each unique electroreceptor, innervated by ~3 afferents from the Anterior Lateral Line cranial nerves [16]. A large electrosensory neuroepithelium (EN), one cell thick and hemiellipsoid-shaped, lines the interior of each AO's basal pole. The EN is composed of only support cells, receptor cells, and afferent terminals [14]. Counts and dimensions of EN cells have been reported in preadult *Polyodon* [16]. The expanded apical faces of support cells comprise ~97% of an EN's apical area. Support cells have hourglass shapes: their pinched-in middle passes through voids between an array of rounded pear-shaped receptor cells. Thus, support cells span the EN, contacting both the AO lumen and the EN's basal lamina [14].

**Table 1. Lectins used.**

| Lectin abbrev. | Lectin name, Species of origin | Inhibiting sugar | Trials | Conjugation |
|---|---|---|---|---|
| ConA | Concanavalin A, *Canavalia ensiformis* | Mannose | 8 | b, CF488A |
| DBA | *Dolichos biforus* agglutinin | GalNAc, D-Galactose | 5 | b |
| LEL | Tomato lectin, *Lycopersicon esculentum* | GlcNAc | 2 | b |
| PNA | Peanut agglutinin, *Arachis hypogaea* | D-Galactose | 9 | b, CF488A |
| RCA$_{120}$ | *Ricinus communis* agglutinin type I | D-Galactose | 9 | b, fluorescein |
| SBA | Soybean agglutinin, *Glycine maximus* | GalNAc, D-Galactose | 6 | b, fluorescein |
| UEA1 | *Ulex europaeus* agglutinin type I | L-Fucose | 2 | b |
| WGA | Wheat germ agglutinin, *Triticum vulgaris* | GlcNAc, Chitin hydrolysate | >10 | b, CF488A |

We report that lectins WGA or SBA bind to the gel of *Polyodon* AOs, supporting homology with elasmobranch AoL, and that gel is secreted mainly by support cells of the large EN at each AO's basal pole. We modeled how secretion, flow, and accelerated exit of gel may defend against invasion of AOs by microbes or small parasites. The carbohydrate-related selectivity of different lectins (Table 1) [17, 18] presumably was the basis of our lectin labeling.

See [17, 18] for summaries of binding site selectivities for different lectins (Discussion). b: Biotinylated lectin, Vector, with streptavidin DyLight488 reporter. CF488A fluor: Biotium. Other fluorescent conjugates of lectins were also tried, with Alexa488 (Thermo), CY3 (Vector), or rhodamine (Vector). "Inhibiting" or "eluting" sugars competitively inhibit binding of lectins to tissue sections or columns of immobilized ligands. GalNAc: N acetyl-D-galactosamine. GlcNAc: N acetyl-D-glucosamine. The number of trials was the number of days when a given lectin was applied to sections of ampullary organs. Additives to buffer for biotinylated lectins: 0.1 mM $CaCl_2$ and $ZnCl_2$.

## Materials and methods

Our use of paddlefish followed protocols approved by the Institutional Animal Care and Use Committee at Ohio University (USA), in protocols L05-23, 13-L-047, or 16-L-020. Paddlefish purchased commercially were maintained in a large recirculating aquarium system. Data were from 8 preadult paddlefish, most ~1 y age, others 2–3 y, of 25–42 cm eye-to-fork length.

### Tissue preparation

Skin on the rostrum was fixed per two different protocols. (i) Brief fixation: Skin on the rostrum was fixed for ~45 min by vascular perfusion and immersion. A paddlefish was deeply anesthetized by injection into red trunk musculature of 2 x 0.2 mL of a 0.5 g/mL saline solution of alfaxalone in a cyclodextrin carrier (Cyclodextrin Technologies Development Inc.). A catheter was ligatured into the conus arteriosus for perfusion of the vasculature at high flow rate (>30 mL/min). Euthanasia was by exsanguination with isotonic PBS containing heparin (anticoagulant) and Na nitroprusside (vasodilator), followed by perfusion fixation for 10–20 min with cooled 4% w/v formaldehyde (from paraformaldehyde, without methanol), 0.25% picric acid, 50 mM NaCl, and 50 mM phosphate buffer, pH 7.5. During perfusion, the rostrum surface was flooded with fixative. After perfusion, soft tissue from the base of the rostrum was quickly cut into ~5x5x5 mm blocks, and immersed in the same fixative solution until a total elapsed time of ~45 min since the start of vascular perfusion. Tissue blocks were then washed in cold PBS with repeated changes until the yellow color of picrate dissipated. After overnight cryoprotection in cold 20% sucrose, a block was oriented in a mold in a 1:2 mixture of OCT gel and 20% sucrose, snap frozen on dry ice / isopentane without immersion, and stored at –30°C until use. Frozen sections were cut on a motorized cryostat (Reichert-Jung 2800E) using disposable blades (Feather), air dried onto amine-adhesive slides, and stored at –30°C.

(ii) Hard fixation: To better preserve microvilli on EN support cells, circular biopsies (7 mm Harris Uni-Core tissue punch) of skin from the rostrum base were fixed by immersion 24 h at 22 ºC [19] in the fixative above. The ~1 mm-thick skin with ampullary organs was delaminated (cut) from subdermal cartilage before fixation. After washing and cryoprotection, a skin disk was frozen and sectioned as above. Slides were rehydrated in 50 mM TRIS buffer, 2 h to overnight, to quench unreacted fixation aldehydes.

### Fluorescent labeling

Table 1 lists the 8 lectins applied to *Polyodon* ERs. Initially a screening kit with biotinylated lectins (Vector BK-1000) was used. A given section was incubated with only a single type of

lectin-biotin conjugate. Slides with 10 or 50 μ cross sections of skin from the rostrum base were brought to room temperature for 10 min, then rehydrated in 50 mM TRIS buffer for 10 min. Slides were blocked using unlabeled streptavidin, then biotin, from a blocking kit (Vector), using Carbo-Free buffer (Vector) with 0.1% v/v Triton X100. The TRIS saline and blocking preincubations also acted to quench free aldehydes from fixation. Biotinylated lectin was applied at 1–5 μg/mL, in Carbo-Free buffer with 0.1 mM $CaCl_2$ and $ZnCl_2$ added, for 30 min at 22 ˚C. Slides were then washed with TRIS buffer containing Triton X100 for 3 x 10 min. To report sites of bound lectin-biotin, streptavidin conjugated to DyLight488 fluor was applied in Carbo-Free buffer at 10 μg/mL for 30 min. Then, sections were washed in TRIS buffer for 3 x 10 min, and coverslipped with a glycerol-based antifade mounting medium (VectaShield). After coverslipping, slides were refrigerated overnight and imaged the next day.

In later work, a lectin conjugated to a green- or red-emitting fluor (Vector or Biotium) was applied singly at 2–65 μg/mL (Table 1). Fluorescence from some conjugates faded rapidly after application, and was photographed within a few hours of coverslipping. PNA-CF488A and $RCA_{120}$-fluorescein yielded high backgrounds. A lectin-fluor conjugate was sometimes mixed with 2˚ secondary Abs or other probes. These included DAPI to stain cell nuclei, or a a red-fluorescent conjugate of phalloidin (Biotium) to reveal F-actin. Phalloidin was dissolved in methanol, then diluted into buffer to 5 μg/mL final concentration.

Indirect immunofluorescent labeling using commercial primary antibodies (Abs) and fluor-conjugated secondary Abs was conventional [16]. The goat secondary Abs (Biotium) were "highly cross-adsorbed" against immunoglobulins of other species. Negative control experiments of omitting primary Ab confirmed low nonspecific binding of secondary Abs. The primary Abs used included monoclonal anti-acetylated-α-tubulin (RRID AB_628409, t > 10, Santa Cruz sc-23950), and polyclonal Abs, most purified by antigen affinity: anti-β-catenin (AB_10681341, t = 4, Bethyl IHC-00584); anti-collagen-I (AB_10721155, t = 6, GeneTex GTX112731); anti-MARVELD2 (tricellulin, t = 2, Proteintech 13515-1-AP); anti-myelin basic protein (MBP, AB_1720890, t > 10, GenScript A01407); anti-neurofilament-H (AB_2313552, t > 10, Aves NFH); anti-parvalbumin-α (AB_1720924, t > 10, GenScript A01439); anti-S100-β (UniProt P04271, t = 1, GeneTex GTX129573); anti-TJP2 (tight junction protein 2, AB_10726662, t = 4, GeneTex GTX103135); or anti-ZO1 (tight junction protein 1, AB_1952257, t > 10, GeneTex GTX108613).

## Imaging

A widefield (digital camera based) Nikon epifluorescence microscope had stepper-motorized focus. It had three single-fluor filter sets with ~50 nm-wide blue, green, or orange/red emission passbands (Chroma Technology). An electrical shutter blocked the Hg arc lamp except when signaled to open in timed exposures by a software-controlled camera. A Diagnostic Instruments model 14.5 monochrome camera with 2048 x 2048 pixels, or a model IN421 Bayer filter RGB color camera with 1600 x 1200 pixels, was operated using SPOT software. Both cameras had a Kodak CCD sensor with 7.4 x 7.4 μ pixels, and built-in anti-blooming. The smaller CCD sensor of the RGB camera avoided optical vignetting in image corners. Nikon objective lenses included 40x 0.7 NA, 60x 1.4 NA, 100x 1.25 NA.

Some images were acquired on a Zeiss LSM510 laser confocal microscope system using Zeiss Zen imaging software.

Colors shown in illustrations match the emission fluorescence color. Images were processed or measured using Adobe Photoshop CS6, Fiji-ImageJ, or Nikon NIS software. Illustrations may show flattened projections from z-stacks of widefield images, from Photoshop's Auto-Blend algorithm [20]. Contrast was usually increased; gamma = 1. Widefield images were

deblurred by Photoshop's Smart Sharpen filter, which uses 2D deconvolution [20]. For confocal images, 2D 'blind' deconvolution with 40 iterations was performed in AutoQuant (Nikon NIS) software. Profile plots in Fiji-ImageJ were from grayscale images.

## Results

*Polyodon* ampullary organs (AOs) are filled with gel [14], as in sturgeons [12] and homologous AoL of elasmobranchs, although the gel in *Polyodon* AOs is difficult to discern. Given the gel's glycoprotein composition in elasmobranchs (INTRODUCTION), we used lectins (Table 1) to label the gel and other components of *Polyodon* electroreceptors, and also accessory structures of surrounding tissues.

We imaged AOs in frozen sections from the base of the rostrum where fish of age ~1 y had less cartilage than anteriorly. Most illustrations show the apical or distal direction (toward the skin surface) as 'up'. Values are reported as mean ± SD (v– 1).

### Lectins WGA or SBA labeled AO gel

Bona fide AO gel was identified as textured loose material in AO lumens, labeled by lectins WGA or SBA, showing irregularities and bright particles, whereas adjacent sites showed low background fluorescence. The inhomogeneity of AO gel contrasted with uniform high background fluorescence obtained with certain lectin-fluor conjugates (below).

**WGA.**  Wheat germ agglutinin from *Triticum vulgaris* binds to dimers and trimers of the amino carbohydrate N-acetyl-D-glucosamine, and also sialic acid groups (DISCUSSION). N-acetyl-D-glucosamine was found at high concentration in canal gel of AoL of marine sharks and rays [4–7].

Figs 1A$_1$ and 2A show WGA-biotin (WGA-b) label of cross sections of briefly fixed whole AOs in *Polyodon* rostrum skin. WGA-b labeled textured loose material in the AO lumens,

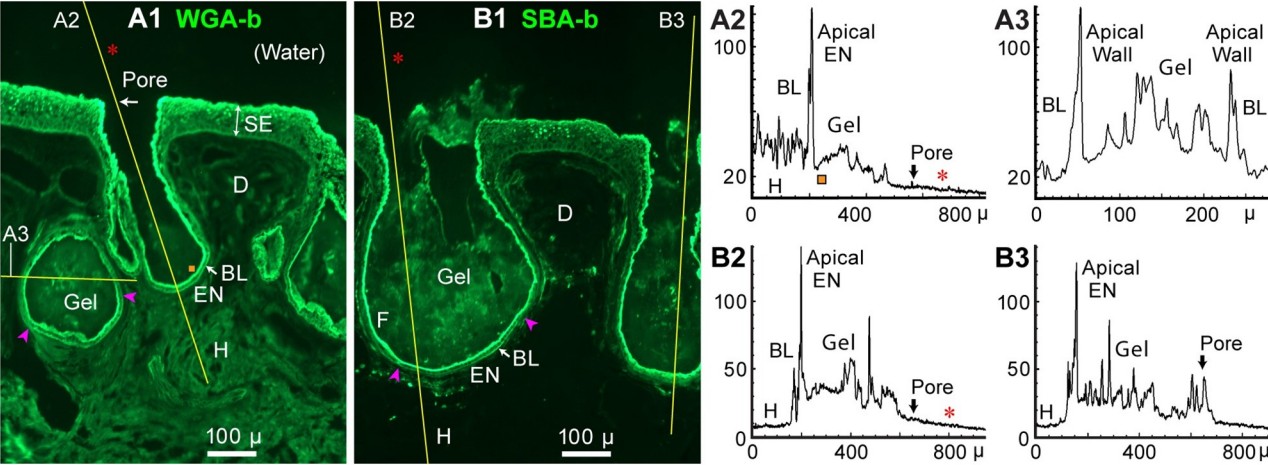

**Fig 1. AO gel label by lectins WGA or SBA; profile plots.** (**A1, B1**) Frozen 10 μ cross sections of *Polyodon* AOs were labeled with biotinylated lectins and streptavidin-DyLight488 fluorescent reporter (Vector). Single raw widefield images, 10x 0.3 NA lens, RGB camera. (**A1**) WGA-biotin (WGA-b), 5 μg/mL. (**B1**) SBA-biotin, 5 μg/mL, preabsorbed with chitin hydrolysate solution. -b: Biotinylated lectin. BL: Basal Lamina. D: Dermis between AOs. EN: Electrosensory neuroepithelium. F: Flattened smooth ampulla wall epithelium, deep in AOs, distal of the EN. H: Hypodermis deep to AOs. Red arrowheads: Limits of an EN. SE: Striated ectoderm; a double arrow marks its thickness. Yellow lines: Paths of profile plots, identified by their panel codes. (**A2**) Profile plot of pixel gray values (8-bit scale) along the left AO in **A1**, and continuing outside its pore (red *). Apical EN: Maximal emission near the apical face of an EN. (**A1, A2**) Orange ■: Gap between gel and EN; see DISCUSSION. (**A3**) Profile plot from left to right across a gel-filled partial AO in **A1**; maximal fluorescence was from the apical face of non-EN epithelia of the ampulla walls. (**B2, B3**) Profile plots from **B1**. SBA+ gel was disturbed in the neck and pore of these AOs.

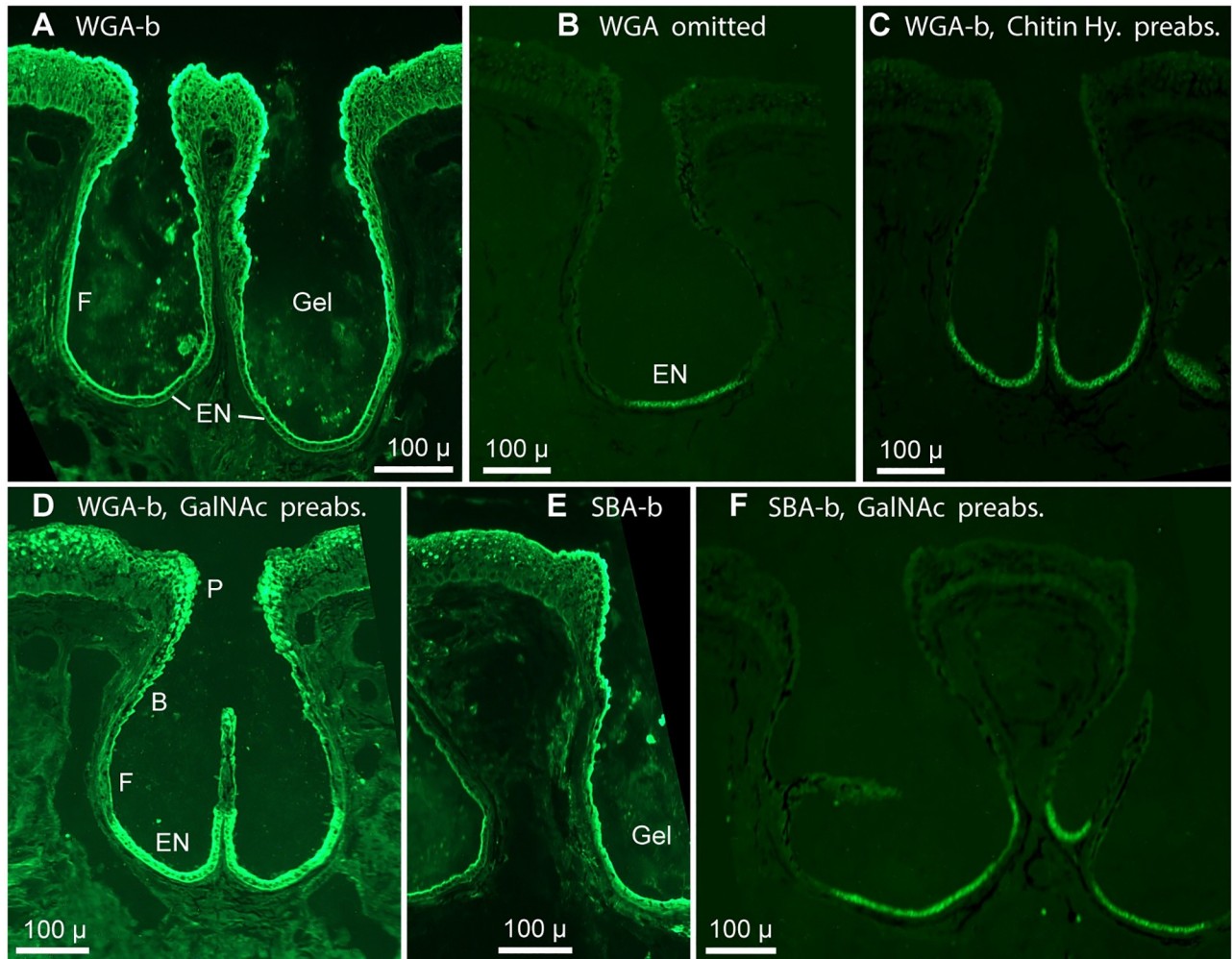

**Fig 2. Controls: Validation of WGA, SBA selectivity.** Lectin-biotin (-b) conjugates, 5 μg/mL final concentration, and streptavidin-DyLight488 reporter were applied. See RESULTS text for preabsorption protocols. All panels show single widefield images of cross sections of AOs; RGB camera, 10x 0.3 NA lens. Similar image processing was applied to all panels, including deblurring and increased contrast. B: Barb-like protrusions from transitional epithelial cells of AO interior wall. EN: Electrosensory neuroepithelium. F: Flattened epithelial cells of the deep interior ampulla wall in AOs. P: Papillas lining the neck and pore. (**A**) WGA-biotin labeled gel in the lumen of AOs, the EN, the interior epithelium of the ampulla wall, neck, and pore, and in the skin's striated ectoderm. (**B**) Background fluorescence was low when WGA was omitted, but reporter was applied. EN: Autofluorescence from EN receptor cells (see S1A Fig). (**C**) Nil label when WGA-biotin was preabsorbed (preabs.) with chitin hydrolysate (Chitin Hy.). (**D**) AO label like **A** when WGA-biotin was preabsorbed with N-acetyl-D-galactosamine (GalNAc). (**E**) SBA-biotin labeled AOs and luminal gel. (**F**) Label by SBA-biotin was abolished, as expected, when it was preabsorbed with N-acetyl-D-galactosamine, its preferred ligand. Also, Fig 1B1 shows labeling of AOs, similar to **E**, by SBA-biotin preabsorbed with chitin hydrolysate.

especially deeper in AOs, identified as AO gel. It showed irregularities and bright particles. Raw image background was low, almost black.

Maximal WGA-b[+] label occurred at the apical surface of the electrosensory neuroepithelium (EN) lining the basal pole of AOs, and at particles in gel. WGA-b also brightly labeled the apical face of the entire epithelium lining an AO, and the skin surface.

A WGA-b[+] basal lamina (BL) surrounded all parts of an AO including the EN. WGA-b labeled moderately the hypodermis under AOs, including pockets of translucent connective tissue (H) [16]. Loose dermis (D) connective tissue, between AOs laterally, was less stained by WGA-b.

In profile plots along the length of an AO, using Fiji-ImageJ software, the WGA-b$^+$ fluorescence from ampulla gel was highest near the EN (proximally), and tended to decline going distally towards the skin pore (Fig 1A$_2$). WGA-b$^+$ label of gel was moderate but well above background fluorescence. A profile plot (Fig 1A$_3$) across the width of an AO, distal of its EN (see **>**), showed obvious WGA-b$^+$ label of AO gel, and maxima at the apical face of non-EN epithelia lining the ampulla walls.

**SBA.** Soybean agglutinin from *Glycine maximus* binds to another type of amino sugar (N-acetyl-D-galactosamine) found at high concentration in canal gel of elasmobranch AoL [4–7]. SBA-biotin also labeled textured gel in *Polyodon* AOs (Gel, Figs 1B$_1$ and 2E). Profile plots of SBA-b$^+$ binding along the length of an AO's lumen (Fig 1B$_2$ and 1B$_3$) typically showed highest gel fluorescence near an EN, and declining gel fluorescence going distally (as for WGA).

Maximal SBA-b label occurred in the EN apically. The apical face of other epithelia throughout an AO's lumen were also brightly SBA-b$^+$. Dermis or hypodermis near AOs were hardly stained by SBA-b (Figs 1B$_1$ and 2E).

Outside a pore, above-background WGA-b$^+$ or SBA-b$^+$ fluorescence persisted (red $^*$, Fig 1A$_2$ and 1B$_2$), but declined rapidly with distance distally, suggesting expelled gel (DISCUSSION).

We inferred that the entire luminal epithelial surfaces of AOs, all lectin-bright, were covered densely by ligands of WGA and SBA.

These data used biotinylated WGA or SBA (Vector). Fluor conjugates of these lectins labeled AOs similarly, including positive label of AO gel (below). Our attempts to label AO gel with other lectins yielded negative or inconclusive results (DISCUSSION).

## Lectin controls

To reduce background when screening with biotinylated lectins, frozen sections of briefly fixed tissue were routinely preabsorbed with TRIS saline, non-fluorescent streptavidin, then biotin (METHODS). This avoided unwanted labeling of mitochondria by streptavidin reporter [21], and quenched possible nonspecific binding of lectins due to free aldehydes remaining from fixation. Confirmation came from a negative control of omitting lectin but applying the fluorescent streptavidin reporter, which yielded low background labeling of AO tissue (Fig 2B). Dim remaining emissions were autofluorescence from EN receptor cells (S1A Fig) and some cells of striated ectoderm.

In preabsorption controls, (i) WGA-biotin solution in CarboFree buffer was mixed 1:1 (v/v) in a tube with chitin hydrolysate solution (Vector SP-0090) composed of N-acetyl-D-glucosamine and oligomers of this amino sugar. The mixture was incubated 1–2 h @ 22°C, then applied to frozen sections of briefly fixed tissue on slides (METHODS). The final concentration of WGA-b was 5 μg/mL. The concentrations of GalNAc and its oligomers in chitin hydrolysate solution were not stated (Vector SP-0090 datasheet), but were in excess. This mixture abolished all binding of WGA-b, including in AO gel, EN, AO wall, connective tissue, and striated ectoderm (Fig 2C vs. 2A). The mixture's high ionic strength was a concern (DISCUSSION). (ii) Similar preabsorption with chitin hydrolysate solution also abolished AO label by lectin LEL-biotin, but (iii) had little effect on AO labeling by SBA-biotin (Figs 1B$_1$ vs. 2E), nor by biotinylated ConA, PNA, or RCA, as expected from their respective carbohydrate selectivities [17, 18]. (iv) Similar preabsorption of SBA-biotin with N-acetyl-D-galactosamine (Vector S-9001; 100 mM final concentration), then application of lectin-GalNAc mixture to skin sections, abolished SBA-b label of *Polyodon* AOs (Fig 2F vs. 2E), as expected. (v) Preabsorption with GalNAc had little effect on AO labeling by biotinylated WGA (Fig 2D vs. 2A), ConA, or RCA, as expected. All biotinylated lectins were at 5 μg/mL (high) final concentration for preabsorption

controls. (vi) We discontinued use of lectin DBA-biotin after finding that preabsorption with GalNAc, its preferred ligand, failed to abolish DBA binding in AOs.

## Lectin label of EN

We found that lectins WGA or SBA, which labeled the AO gel (above), extensively labeled EN support cells (§1–3 here), implicating support cells as a major source of the gel filling the lumens of AOs [12]. We assigned whether lectin label was on receptor or support cells based on the more apical position of support cell nuclei [15].

**1. WGA+ apical surface label of EN support cells.** Conjugate WGA-CF488A labeled brightly the apical face of EN, in parallel (en face) views, but bound to only support cells (S, Fig 3A1). Peaks in surface profile plots (Fig 3A2 and 3A3), presumably corresponded to abundant long microvilli reported on support cells [9, 12, 14, 22, 23], including at their edges. The relatively constant minimum intensity values (> on y axis, Fig 3A3) likely corresponded to apical surface label per se on support cells, apart from microvilli.

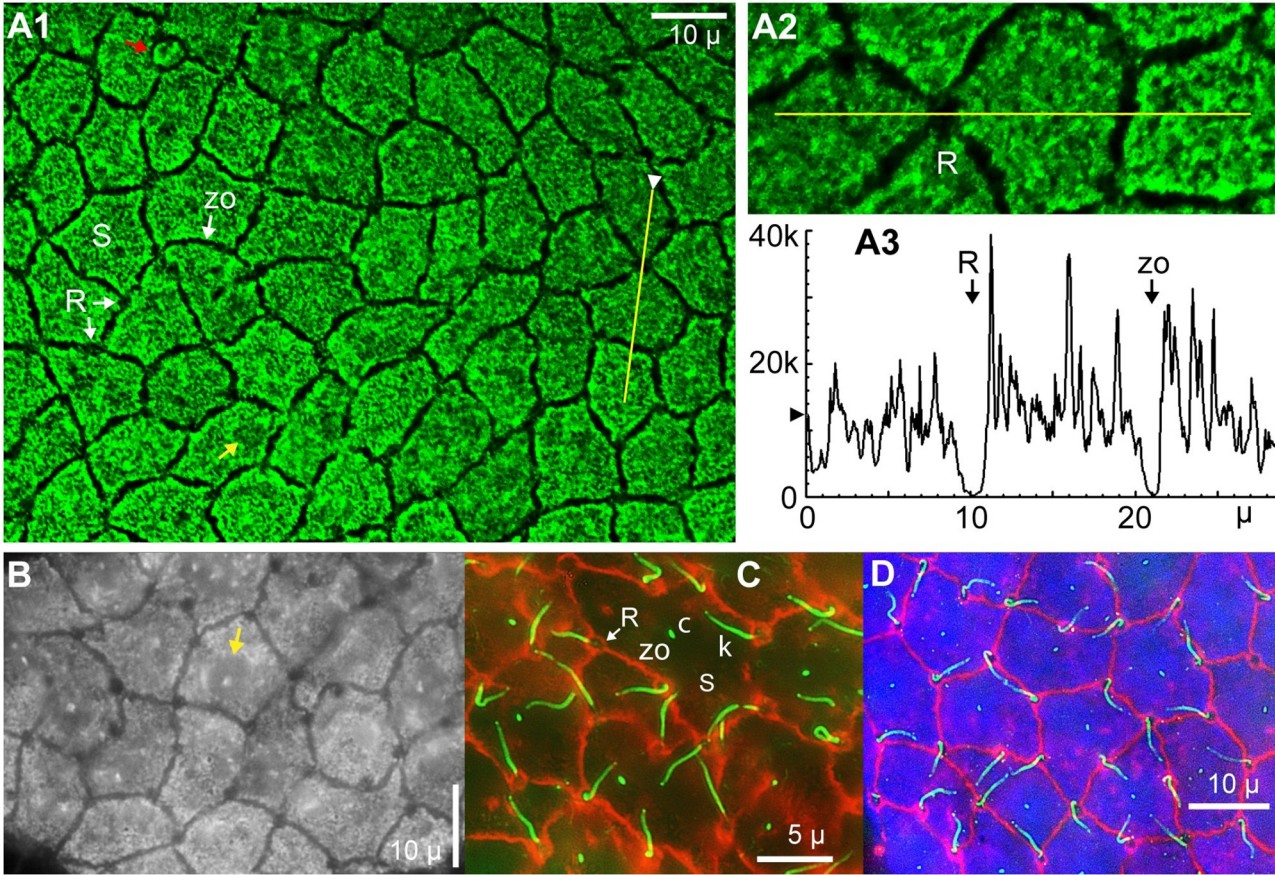

**Fig 3. Apical surface views of WGA-labeled EN.** Parallel (en face) views of the apical surface of partial EN, in briefly fixed AOs. (**A1**) WGA-CF488A conjugate (Biotium) labeled the apical faces of support cells (S). R: Receptor cell. zo: Zona occludens. Red arrow: Support cell with a small-area apical face. Confocal single image, deconvolved, 63x 1.4 NA lens. The 'black' end of this image's gray scale was unmodified. (**A2, A3**) Profile plot across part of **A1**. Fluorescence emissions reached zero at a receptor cell's apical face (R) or a zona occludens (zo), indicating non-label by WGA. (**B–D**) Widefield stack images. (**B**) WGA-CF488A label; 60x lens, monochrome camera. Yellow arrow: Region of a support cell's cilium. A similar zone is marked in **A1**, yellow arrow. (**C**) Phalloidin (orange) labeled the zona occludens (zo) joining EN support (S) and receptor (R) cells apically. One kinocilium (k) per receptor cell was labeled by anti-acetylated-α-tubulin (green), which also labeled short cilia (c) on support cells. Monochrome camera, 100x lens, 0.5 μ stack steps. (**D**) Triple label with WGA-CF488A (blue), phalloidin (red), and anti-acetylated-α-tubulin (green). Monochrome camera, 60x lens, 0.4 μ stack steps.

Features not labeled by WGA, showing zero emission in profile plots (Fig 3A₃), included (i) the prominent zona occludens (zo) junctions surrounding support and receptor cells apically, and (ii) the circular ~1.5 μ diameter apical face of receptor cells (R, Fig 3A₁). Some profile plots across a receptor cell showed a weak central feature.

Receptor cells have a co-localized kinocilium, labeled by anti-acetylated-α-tubulin (k, Fig 3C). This antibody also revealed a short cilium on EN support cells (c, Fig 3C), centered in a specialized region (yellow arrow, Fig 3A₁ and 3B). Fig 3D shows these components co-labeled.

The apical surface of EN was sometimes flat (as Fig 3A), but typically was corrugated over a 1–3 μ vertical range by dome-like elevations over the nuclei of some support cells, visible in cross sections (Fig 4).

**2. Apical microvilli on support cells.** Several lectins brightly labeled numerous long microvilli on the apical faces of support cells [9, 12, 14, 22, 23], imaged in cross sections of EN from hard-fixed skin of the rostrum (METHODS) using lectin-fluor conjugates. The apical microvilli (μv, Fig 4A₄) were phalloidin⁺ [9]. Microvilli were labeled maximally by lectins WGA (Fig 4A), SBA (Fig 4D and inset), or PNA (Fig 5A), and faintly by RCA₁₂₀ (Fig 5D). The DISCUSSION considers whether a lectin's label of microvilli may correlate with label of luminal gel. The apical faces of receptor cells had no or few WGA⁺ (Fig 3) or SBA⁺ (arrow, R, Fig 4D) microvilli.

In briefly fixed EN, collapsed lectin⁺ microvilli likely formed the brightest parts of the WGA⁺/SBA⁺ apical layer on support cells (Fig 4B and 4E), or microvilli were lost.

**3. Lectin⁺ vesicles or organelles in EN support cells.** Letters in Fig 4A₂ and 4E mark examples of nuclei (n) or cytoplasm (c) of support (s) or receptor (r) cells, and apical microvilli (μv), in cross sections of EN. We estimated lectin label of EN cells based on the apparent brightness of fluorescence in a lectin's color channel (DISCUSSION), relative to background. The cytoplasm of receptor cells was autofluorescent (S1A Fig) (below).

**WGA** strongly labeled numerous vesicles and organelles in the apical halves of support cells, using either a fluor conjugate (Fig 4A₁) or WGA-biotin (Fig 4B). There, apparent brightness increased going apically. A thin sheet of cytoplasm distal of the nucleus was brightly labeled, but less brightly than in juxta-luminal microvilli (DISCUSSION). This is seen in the support cell marked 'S' in Fig 4A₂, imaged across the EN. WGA⁺ label continued into the basal pedicels of support cells, adjoining the basal lamina [14].

**SBA**-biotin yielded superior label of support cell cytoplasm in briefly fixed EN, which increased in apparent brightness apically (Fig 4E) (DISCUSSION). SBA-fluorescein yielded elevated background (Fig 4D) in hard-fixed EN, but did reveal the gel filling AOs (Fig 4C), like SBA-biotin (Fig 1B).

**ConA:** A different population of smaller and more uniform vesicles in EN support cells was labeled by lectin ConA (Fig 5B and 5C), filling all cell regions including pedicels (p). These ConA⁺ vesicles were likely non-secretory. ConA did not label microvilli, leaving a void at the expected apical sites (a, Fig 5B). ConA-biotin or ConA-CF488A labeled EN similarly, for both fixation protocols.

**RCA₁₂₀:** Numerous tubules, ~1 μ across, were brightly labeled apically in EN by RCA₁₂₀-fluorescein conjugate in hard-fixed AOs. The RCA⁺ tubules occurred at the level of EN nuclei (Fig 5D and 5E), and formed lateral nests centered on nuclei in parallel views of EN (Fig 5F). These were likely Golgi tubules, which are known targets of RCA₁₂₀ binding [24]. Some labeled tubules were distal of support cell nuclei (G, Fig 5D), that is, definitely in support cells.

RCA₁₂₀ was distinctive in labeling crossed layers of thick coiled elastic fibers in a ~15 μ -thick layer of connective tissue under EN exteriorly (f, Fig 5D and 5E), joined to the AO wall.

The prominent basal lamina of EN, which was WGA⁺ and SBA⁺ (above), was usually not labeled by PNA (Fig 5A), ConA (Fig 5B and 5C), or RCA₁₂₀ (Fig 5D).

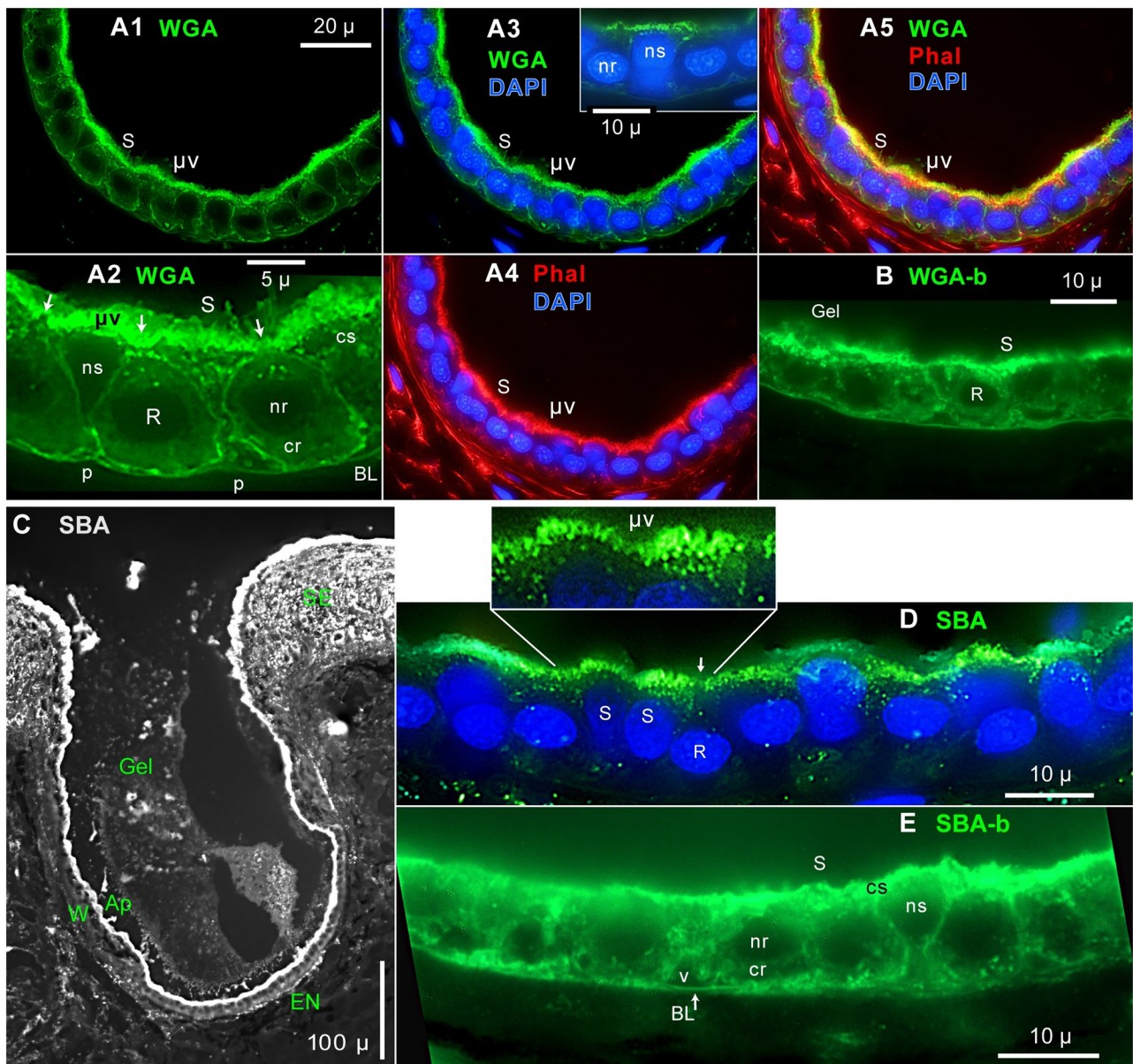

**Fig 4. WGA / SBA label in electrosensory neuroepithelia.** Transverse cross sections (10 µ) of EN were labeled with lectins WGA (**A, B**) or SBA (**C–E**). Tissue fixation or lectin conjugation were varied. Images show flattened projections from widefield partial stacks, with an EN's apical face 'up'. The nucleus (n) or cytoplasm (c) of support (s) or receptor (r) cells are indicated. Nuclei of support cells are more apical than nuclei of receptor cells [15]. BL: Basal lamina. L: Lumen of AO. p: Pedicel of support cell [14]. v: Void containing afferent branches [16]. (**A, C, D**) 'Hard' fixation for 24 h preserved the apical microvilli (µv) on EN support cells, labeled using lectin-fluor conjugates; monochrome camera. (**B, E**) Brief (45 min) fixation, and label with lectin-biotin (-b) conjugates using streptavidin-DyLight488 reporter; RGB camera. (**A1–A5**) Triple label with WGA-CF488A (green, Biotium, 2 µg/ mL), and phalloidin-CF594 (Phal, red, Biotium), and DAPI (blue). (**A1–3**) WGA label. (**A2**) Enlarged part of **A1**, resampled 4x. A support cell (S) was also marked on other panels of **A**. Arrows: Apical face of receptor cell (R). (**A3**) WGA label with DAPI+ nuclei. Inset: Local patch of WGA+ microvilli in a single image from another section, resampled 4x. (**A4**) Phalloidin (Phal) labeled apical microvilli (µv) on support cells. (**A5**) All 3 labels were superimposed. Yellow color of apical microvilli on support cells indicated co-label by WGA and phalloidin. (**B**) WGA-biotin (WGA-b) label of EN, 1 µg/mL; stack of 6 images. (**C, D**) SBA-fluorescein label (Vector, 65 µg/mL). (**C**) This ampullary organ, its gel, and skin were labeled by SBA, which labeled the EN more brightly than in ampulla wall (W), and labeled the apical (Ap) surface of all AO epithelia. SE: Striated ectoderm. (**D**) SBA label of EN. Inset: Expanded view of SBA+ apical microvilli on support cells. (**E**) SBA-biotin label of EN, 5 µg/mL; stack of 3 images. (**A, D**) 60x lens, 0.5 µ stack steps. (**B, E**) 100x lens, 1 µ stack steps. (**C**) 10x 0.45 NA lens.

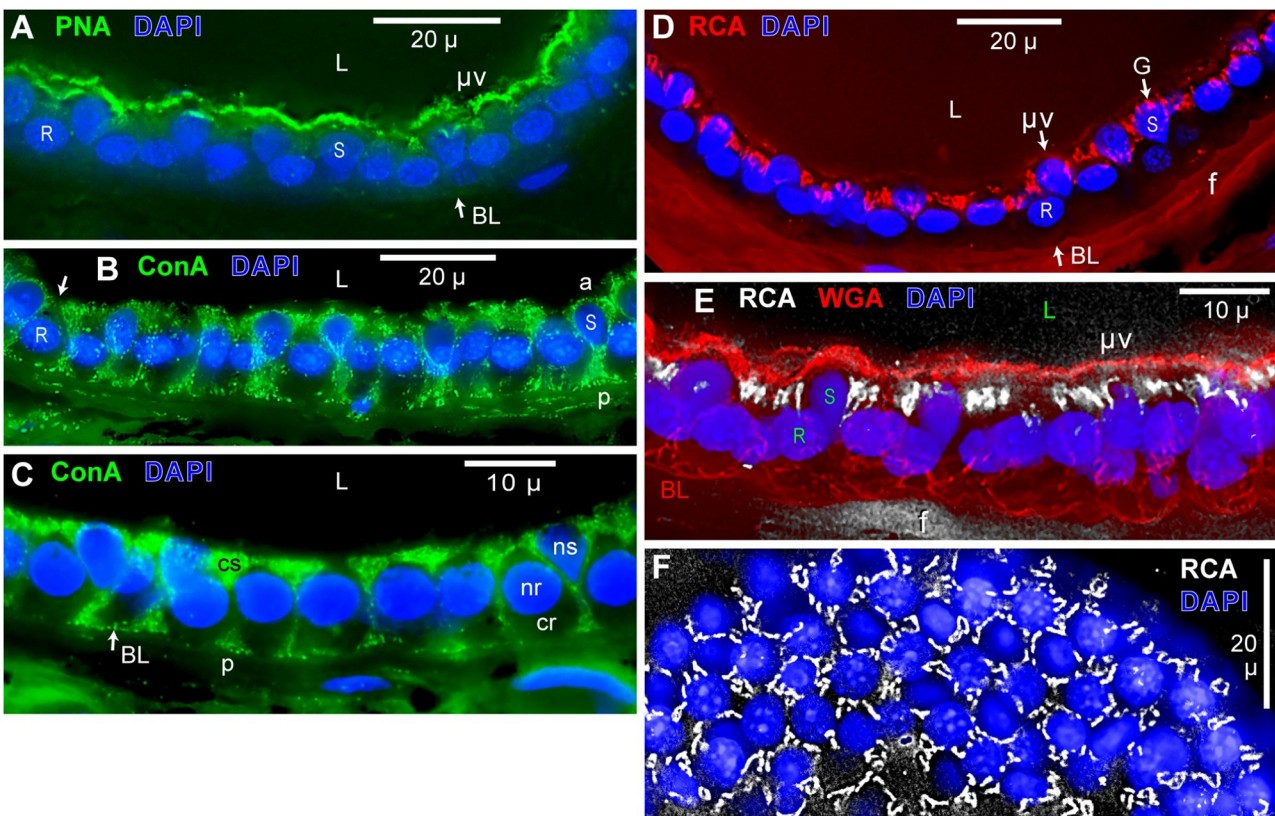

**Fig 5. Lectins PNA, RCA$_{120}$, or ConA labeled EN support cells.** (A–E) Lectin-fluor conjugates were applied to 10 μ transverse sections of EN in hard-fixed skin from the rostrum; see keys. Monochrome camera, 60x lens, 0.5 μ stack steps. (**A**) Lectin PNA-CF488A (Biotium, 48 μg/mL) labeled apical microvilli (μv) on support cells. BL: Expected site of basal lamina. L: Lumen of AO. R: Receptor cell nucleus. S: Support cell nucleus. (**B, C**) Lectin ConA-CF488A (Biotium, 48 μg/mL) labeled small vesicles throughout support cells, in hard-fixed (**B**) or briefly fixed (**C**) EN. a: Apical face of support cell. Arrow: Apical face of receptor cell. Nuclei (n) or cytoplasm (c) of support (s) or receptor (r) cells are labeled. p: Basal pedicel of support cell. (**C**) This EN was thicker than usual. (**D–F**) Lectin RCA120-fluorescein (Vector, 10 μg/mL, without added CaCl$_2$ or ZnCl$_2$), labeled Golgi tubules (G) in the apical half of support cells, and faintly labeled apical microvilli (μv). f: Fibrous RCA+ sub-EN layer. (**F**) En face view of EN, showing DAPI+ nuclei at the center of nests of RCA+ tubules (white).

**4. Glycocalyx on receptor cells?.**  Lectins have been used to reveal exterior glycocalyces surrounding neurons [25–27]. Our attempts to label the exteriors of EN cells using lectins were inconclusive. A difficulty was the autofluorescence of receptor cell cytoplasm at all visible wavelengths. We imaged autofluorescence in unlabeled sections, in enface optical sections within EN, as narrow circular annuli (cr, S1A Fig), corresponding to shells of cytoplasm around nuclei of rounded receptor cells. Small voids in the EN array were the pinched-in waists of hourglass-shaped support cells, which had low or nil autofluorescence.

In 'experimental' data, the cryostat blade sometimes dislodged receptor cells, which are compact and rounded. In images parallel to EN, this revealed circular fluorescent domains ('sockets', green in Fig 6), among arrays of remaining receptor cells whose cytoplasms were labeled by anti-parvalbumin-α (orange in Fig 6B) [16]. For example, ▲ in Fig 6 marked one of the sockets without parvalbumin+ labeling, whereas * marked another s'ocket aligned with a PARV+ receptor cell. Hence the parvalbumin-α+ cytoplasm of a receptor cell was mechanically separable from an aligned extracellular domain surrounding it basolaterally, that could not be attributed to cytoplasmic autofluorescence. We imaged en face three such examples of dis-lodged receptor cells: all showed empty sockets.

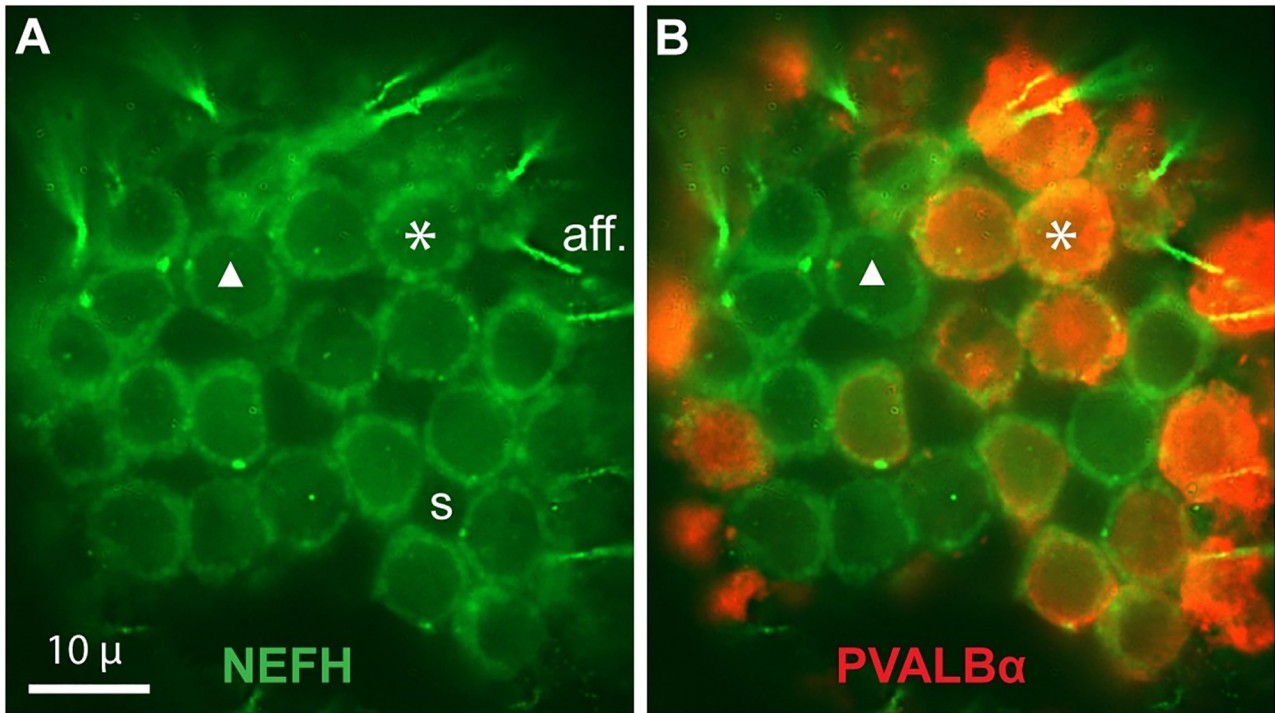

**Fig 6. Extracellular sockets around receptor cells.** (**A, B**) En face stack images of EN receptor cell arrays. Some receptor cells were dislodged by the cryostat blade during sectioning, leaving green-fluorescent 'sockets'. * or ▲: Corresponding sites in **A** and **B**; see Results text. RGB camera, 60x lens. (**A**) Green: Anti-neurofilament-H (NEFH) labeled afferent (aff.) terminals. S: Void corresponding to a support cell. (**B**) Superimposed anti-parvalbumin-α (PVALBα, orange) label of remaining receptor cells.

These extracellular 'socket' domains suggested a basolateral glycocalyx surrounding each receptor cell of *Polyodon* EN, as for photoreceptors of vertebrate retinas [28]. Sockets often had scalloped edges. Some of these ~1 u domains around the perimeter of a RC may be synaptic boutons, as in * Fig 6 which resembles EM of ribbon synapses [14].

## Ampulla wall epithelium

We asked whether additional gel might be secreted from the non-EN interior wall of AOs, which had a larger surface area than the EN, and is lined by a one-cell-thick cuboidal epithelium ('wall cells'). Little has been reported about AO wall cells. We labeled the wall epithelial cells with lectins, including with WGA or SBA which were effective for labeling the AO gel and the gel secretory apparatus of EN support cells (above).

**Morphology.** Epithelial cells of an interior ampulla wall varied along an AO's length. Figs 2D and 7B–7D show a proximal-to-distal sequence of flattened (F), barbed (B), or papilla (P) cells of an AO's luminal epithelium. Deep in an AO, wall cells had relatively smooth and flattened (F) apical faces (Figs $1B_1$, 2A and 2D); these were well-seen in Fig 7C due to the tilted EN. Starting near mid-AO levels, raised barb-like projections, angled toward the pore, were observed on wall epithelial cells (B, Figs 2D and 7B–7D; *, Fig 7H; inset 1, Fig 7I), and continued to near the neck. Larger papillas (P) lined the neck and pore (below). The distal placement of wall projections (barbs, papillas), near and in the neck and pore, was consistent with defensive functions (Discussion).

The thickness of the ampulla wall epithelium, including its basal lamina, was 14.3 ± 1.6 μ, similar to the 13.4 ± 1.9 μ thickness of EN (Tab 1 in S1 Table).

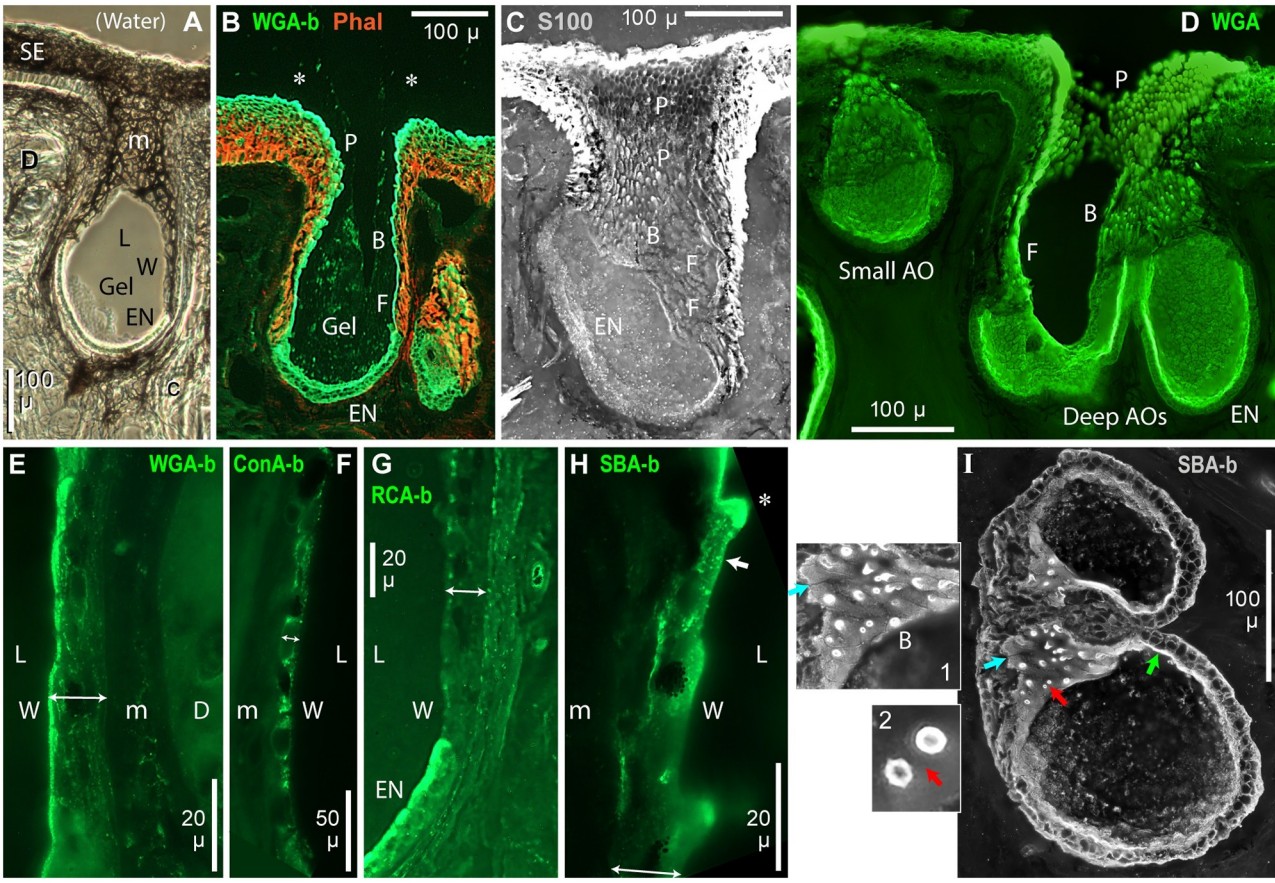

**Fig 7. Cross sections of ampulla wall.** b: Biotinylated lectin. B: Barbed wall cells. c: Capillaries near AO. D: Dermis. EN: Electrosensory neuroepithelium. F: Flattened cells of deep ampulla wall. L: Lumen of AO. m: Melanin-containing outer fibrous layer on AOs. P: Papilla cells lining neck and pore. Phal: Phalloidin label. SE: Striated ectoderm. W: Non-EN ampulla wall epithelium. (**A**) Brightfield cross-section of AO; unstained, 10x phase lens. Partial luminal gel remained at the EN. (**B**) Cross section of an AO and adjacent skin stained with phalloidin (orange) and WGA-biotin (green). *: AO gel with small particles was visible outside the pore. (**C, D**) Parallel views of the interior wall of AOs labeled with anti-S100-ß (**C**) or WGA-CF488A (**D**), showing different cells types of the ampulla wall epithelium, including flattened cells (F) deep in AOs, transitional cells with barb-like protrusions (B), and papilla cells (P) in the neck and pore. (**E–H**) Closeup images of cross sections of ampulla wall showing vesicles in wall epithelial cells labeled by a lectin; see keys. Double arrows: Thickness of wall epithelium. (**E**) WGA-b, 1 μg/mL. (**F**) ConA-b, 5 μg/mL. (**G**) RCA-b, 2.5 μg/mL. (**H**) SBA-b, 5 μg/mL. Arrow: Apical vesicle labeled by SBA-biotin. *: Barb-like SBA+ protrusion from wall cell. (**E, H**) 100x lens. (**F, G**) 40x lens. (**I**) SBA-biotin label of horizontally sectioned ampulla walls; stack projection of a 50 μ section parallel to skin. Blue arrow: En face view of SBA-b+ apical surface labeling on epithelial cells, in a tilted part of the ampulla wall, enlarged in inset 1. Green arrow: SBA-b+ apical label of cross-sectioned epithelial cells of ampulla wall. Red arrow: Sectioned SBA-b+ barb-like protrusions from wall cells, enlarged in inset 2 (resampled 4x).

Outside the wall epithelium was a thick layer of fibrous connective tissue with abundant melanin (m, Fig 7A, 7E, 7F and 7H), and other DAPI[+] cells, forming an outer AO sheath. Numerous capillaries surrounded AOs (c, Fig 7A). Phalloidin (binding to F-actin) strongly stained subsurface fibrous structures of the AO wall and skin (Fig 7B), whereas WGA stained nearer their surfaces. The luminal faces of ampulla walls also had fibrous melanin (W, Fig 7A), unlike the EN which usually lacked apical melanin.

We labeled a distinct border (b, Fig 8A and 8G) at the transition from EN to wall epithelium, in en face views. In cross sections, the transition formed a thickened band in some AOs, but often the transition was continuous (*, Fig 9A and 9B₁), or formed a corner.

**Junction immunolabel.** We imaged intercellular junctions between ampulla wall epithelial cells in images parallel to their luminal surfaces; Fig 8 shows flattened projections of image stacks (METHODS). Anti-ZO1 (Fig 8A–8C) or anti-TJP2 (Fig 8E) labeled tight junctions at cell

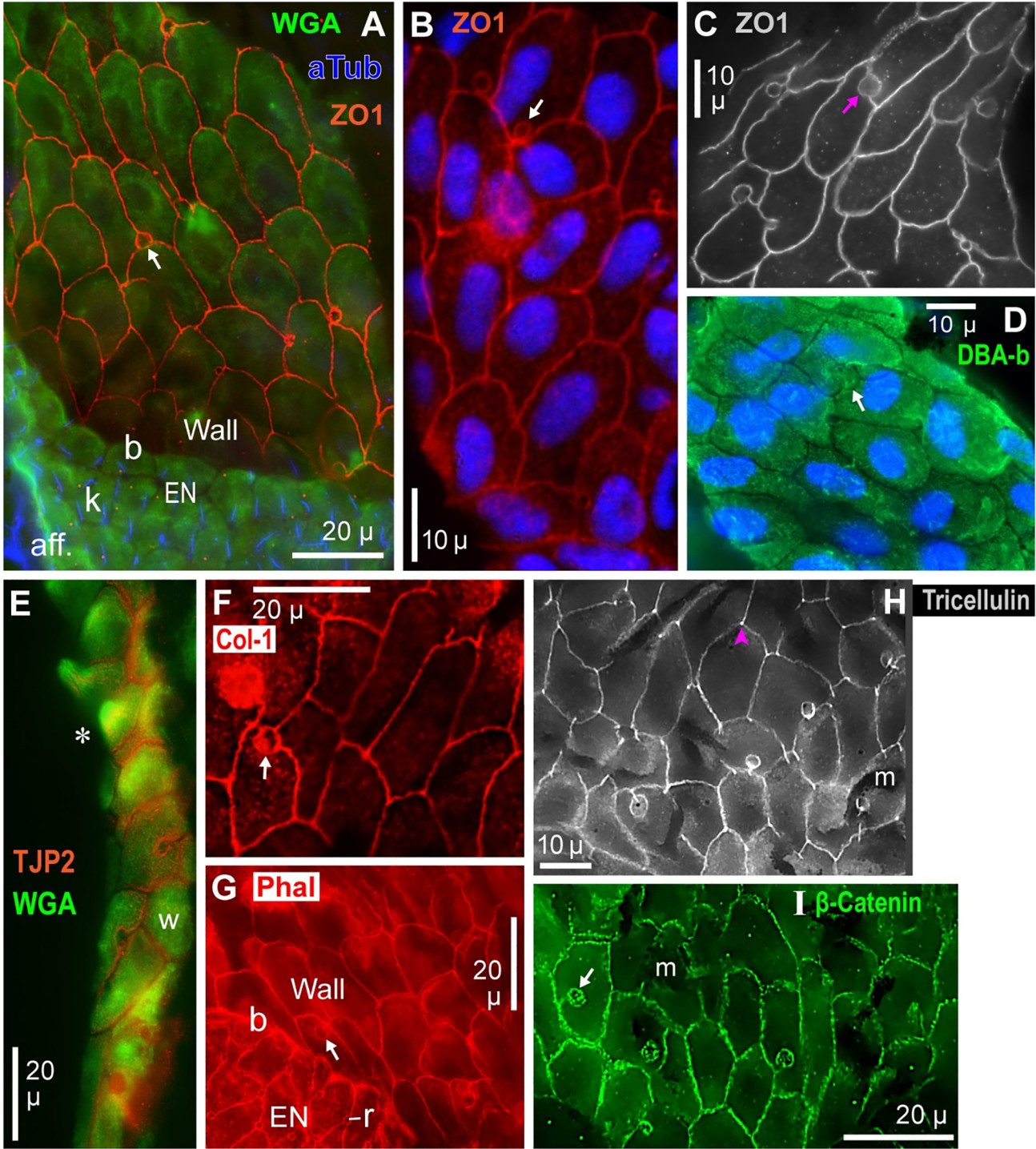

**Fig 8. Ampulla wall cells: Junction immunolabel and lectin surface label.** (**A–I**) Parallel views of the apical faces of ampulla wall epithelial cells. See keys and METHODS for the probes applied; 40x lens. aff: Afferent branches to EN, labeled by anti-acetylated-α-tubulin (aTub). Arrows: Small circular whorls. b: Border between EN and ampulla wall. Col-1: Anti-collagen-1. EN: Electrosensory neuroepithelium. k: Kinocilium labeled by aTub, on each receptor cell's apical face. Phal: Phalloidin. r: Receptor cell apical face. TJP2: Anti-tight junction protein 2. WGA: WGA-CF488A (Biotium) conjugate. ZO1: Anti-tight junction protein 1. (**B, D**) DAPI stain of wall cell nuclei. (**D**) Lectin DBA-biotin bound to the apical surfaces of wall cells but not zona occludens between cells. (**E**) Section of an ampulla's wall; its luminal edge showed barb-like protrusions (*) from wall epithelial cells. (**H**) Anti-tricellulin (MARVELD2) preferentially labeled vertices (arrowhead), where borders of wall cells intersected. (**I**) Anti-β-catenin labeled series of small punctate sites along cell borders. (**G–I**) Strands of dark melanin (m) overlay the wall epithelium.

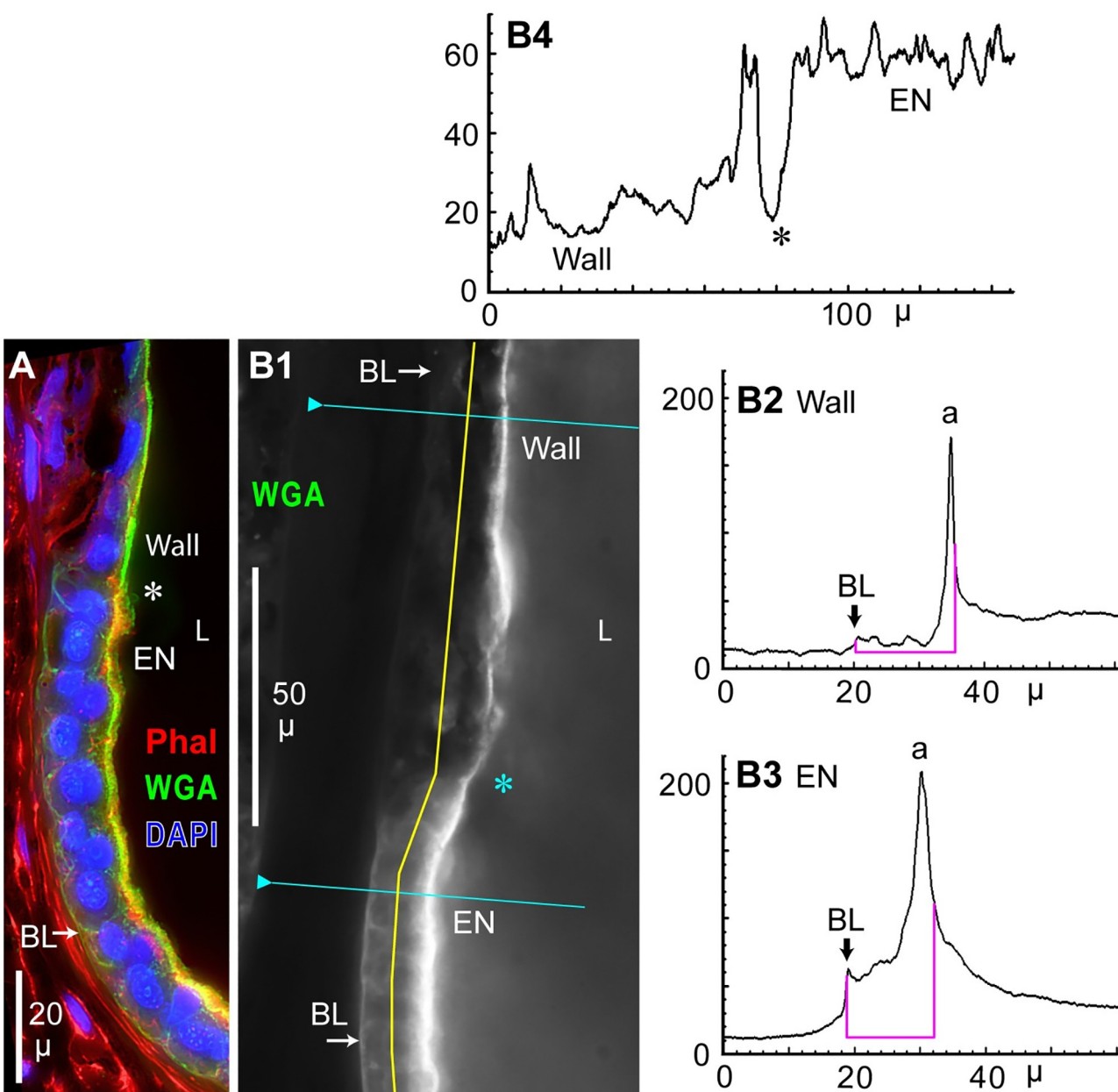

**Fig 9. Comparison of ampulla wall vs. EN label by WGA.** (**A, B1**) Cross sections showing the transition (*) from EN to ampulla wall. (**A**) Triple-label as in Fig 4A5; see keys. The ampulla wall appeared green whereas the EN appeared yellow; see RESULTS text. (**B1**) Green emissions from WGA-CF488A (Biotium) label; brightened deblurred image, monochrome camera, 40x lens. BL: Basal lamina. L: Lumen of AO. (**B2, B3**) Transverse profile plots along the blue lines in **B1**, crossing the wall epithelium (**B2**) or a support cell of the EN (**B3**). These plots were scaled identically, and were from the original raw image of **B1** converted to 8-bit grayscale. Area under the curve was summed between the pink drop lines; see RESULTS text. a: Apical face of epithelium. (**B4**) Lengthwise profile plot along the middle of the wall epithelium and EN (yellow line, **B1**).

borders, consistent with zona occludens. Also, anti-tricellulin (MARVELD2) (Fig 8H) labeled vertices where ~3 borders intersected, as expected for tight junctions [29]. Cell borders were labeled in a continuous manner by anti-collagen-1 (Fig 8F) or phalloidin (Fig 8G), but in a punctate manner by anti-β-catenin (Fig 8I).

The borders around wall cells varied greatly in 2D shape. Wall cell borders usually showed vertices, where borders of adjacent cells met, but borders rarely formed regular polygons, e.g.,

with 5 or 6 sides. Some wall cells had curved borders (Fig 8C and 8G). Their widths or lengths were approx. 10–30 μ; the larger cells were elongated.

Circular whorls (arrows, Fig 8), of unknown origin, were commonly observed among wall cells. Whorls were anucleate (DAPI⁻). They had widths of 3.4 ± 1.3 μ (range 1.8–7.3 μ, v = 32; Tab 2 in S1 Table). Their borders were labeled by all the probes listed above, consistent with zona occludens. Whorls occurred at vertices or along borders of wall cells, or as a cul de sac within the borders of a wall cell (Fig 8). Whorls resembled defects (vortices) associated with curvature of other planar surfaces [30].

**Lectin label of ampulla wall cells.**

i. The apical surfaces of ampulla wall epithelial cells were labeled by lectins SBA-b (inset 1, Fig 7I), WGA-CF488A (Fig 8A), or DBA-b (Fig 8D).

ii. Sparse apical vesicles in the interiors of wall cells were labeled by biotinylated lectins WGA (Fig 7E), SBA (arrow, Fig 7H), or RCA (Fig 7G). These lectin⁺ vesicles occurred near the apical faces, less basally. By contrast, ConA-b⁺ vesicles were dispersed in wall cells (Fig 7F). The level of expression of lectin⁺ cytoplasmic vesicles or organelles was much less in wall epithelial cells compared to EN support cells. Examples included Fig 7G for RCA-biotin, or Fig 7E (wall) vs. Fig 4B (EN) for WGA-biotin at low concentration (1 μg/mL for both images).

iii. The tips of barbs on distal wall cells were brightly labeled by WGA (*, Fig 8E) or SBA-b (*, Fig 7H and inset 1, Fig 7I). The SBA label was apical, as seen in sectioned barbs (inset 2, Fig 7I). Such labeling suggested secretions from barbs (DISCUSSION).

iv. Emissions of wall cells vs. EN support cells were quantitated and compared at a WGA-labeled transition from wall to EN (Fig 9B₁). Profile plots across the wall epithelium or EN (Fig 9B₂ and 9B₃, scaled identically) were along the blue lines in Fig 9B₁. Both plots showed sharp peaks (a) near the apical surfaces. The largest difference between these plots was in the area under the curve. The path gray values (baseline subtracted) were summed over an epithelium's 13–15 μ thickness (between the pink drop lines in Fig 9B₂ and 9B₃), from the basal lamina (BL) to 50% decline from the apical peak. The plot across an EN support cell yielded ~3.6x greater summed fluorescence than the wall cell plot in this example (Tab 4.1 in S1 Table).

For v = 3 examples, trans-EN plots yielded 4.2 ± 0.8 greater summed fluorescence than from trans-wall cell plots (Tab 4.0 in S1 Table). Each path across an EN followed a support cell's cytoplasm. This result applies to the deepest wall cells, near the wall-EN transition, lacking WGA⁺/SBA⁺ barbs.

v. In another comparison, a lengthwise mid-epithelial plot (Fig 9B₄) along the yellow line in Fig 9B₁ showed a ~3x step increase in WGA⁺ fluorescence at the transition (*) from wall epithelium to EN. This difference was visible in Fig 9B₁.

vi. Similarly, SBA⁺ label was brighter in EN than ampulla wall epithelium (Fig 4C).

vii. The dense apical microvilli on EN support cells appeared yellow and thick in Fig 9A due to superimposed label by WGA (green label) and phalloidin (red label), as in Fig 4A₅. By contrast, the apical surface of ampulla wall ('Wall') appeared green and thin, as evidence against phalloidin⁺ microvilli on ampulla wall epithelial cells.

We concluded that non-EN epithelial cells of ampulla wall appeared to be mainly structural or defensive in *Polyodon* AOs (DISCUSSION).

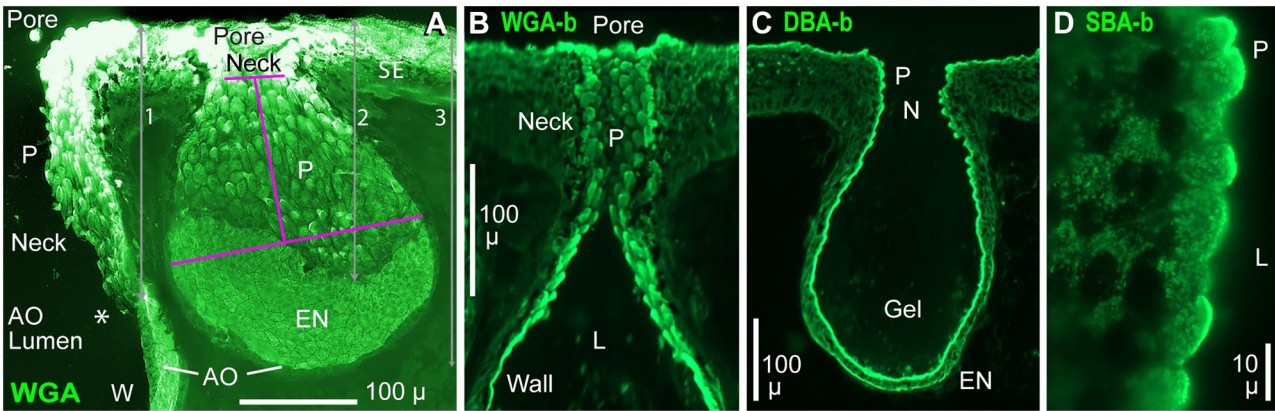

**Fig 10. Lectin-labeled papilla cells lined the AO neck.** (**A**) Cross section of skin showing WGA-CF488A label in the neck and pore of AOs, including a deep AO (left) and an adjacent small AO; 20x 0.5 NA lens. P: Papilla cells covered a ~250 µ-deep band in the neck and pore. *: Transition from papilla cells to flattened ampulla wall cells (W). SE: Striated ectoderm. Small AO: En face view of the interior apical surface of its electrosensory neuroepithelium (EN) and ampulla wall, which included papilla cells (P). Arrow 1: 236 µ depth of the papilla cell band in the deep AO (left side). Arrow 2: 223 µ maximal wall depth of the small AO. Arrow 3: 286 µ total depth of the small AO. Pink construction lines show dimensions of the small AO's tapered ampulla wall, including its height (142 µ), maximum internal width (218 µ), and the neck's ID (51 µ), for a model of gel acceleration (Tab 1 in S2 Table). (**B–D**) Biotinylated (-b) lectins and streptavidin-DyLight488 reporter were used. (**B**) Stack image parallel to the neck interior of an AO, showing WGA-b+ papillas (P) in the neck (N), and that large WGA-b+ papillas, almost parallel to the wall, continued into the distal ampulla; 10x lens. L: Lumen of AO. (**C**) Lectin DBA-b (preabsorbed with chitin hydrolysate solution) labeled the interior epithelium of this AO, including papilla cells (P) in the neck (N). DBA-b+ gel was visible in the AO's lumen; 10x lens. (**D**) Cross section of short papilla cells (P) labeled by SBA-b in the neck / pore of an AO; single image, 100x lens.

## Papilla cells line AO necks

A band of numerous elongated papillas (P, Fig 10A–10D) lined the neck and skin pore of each AO [31]. Papillas projected into the neck's lumen, and were angled toward the pore. The exteriors of papillas were labeled by lectins including WGA (Fig 10A and 10B), DBA-b (Fig 10C), or SBA-b (Fig 10D). The apparent lengths of most WGA+ papillas were 6–21 µ. Papillas varied in morphology within an AO's neck, from short, rounded, and discrete near a pore, to wide, pointed, and partly embedded in the deeper neck. Papillas appeared to join onto (that is, be extensions of) cells making up the cuboidal epithelium lining necks and pores [31].

Papillas formed a continuous toroidal band, to ~250 µ below the rim of skin pores (arrow 1, Fig 10A). In small shallow AOs, the ampulla wall had papillas covering to 200+ µ depth below the skin surface (Fig 10A, arrow 2).

At the border of neck to ampulla wall, papillas transitioned into smaller barb-like apical protrusions of wall cells (above; B, Figs 2D and 7B–7D), usually. Some AOs showed an abrupt change to flattened ampulla wall cells (Fig 10A), but other AOs had papillas continuing into the distal ampulla (Fig 10B).

The functions of the papillas in AO necks and pores are unknown. They may enact defenses against invasions into the AO lumen by pathogens or small parasites (DISCUSSION).

## Lectin label of axon, nerve, glia sheaths

Most lectins we screened were ineffective for labeling individual myelinated axons. Lectin PNA-biotin brightly labeled a thin external sheath on large myelinated ALLn afferent axons to AOs (Fig 11A). This PNA+ sheath continued across nodes (N) and internodes (IN). Such transnodal sheaths of myelinated axons presumably derive from the basal lamina of Schwann wells, with components from endoneurial fibroblasts [32, 33].

PNA also labeled a basal lamina (externally) on large satellite (terminal) glia which ensheath the unmyelinated projections of ALLn afferent terminals to AOs [16].

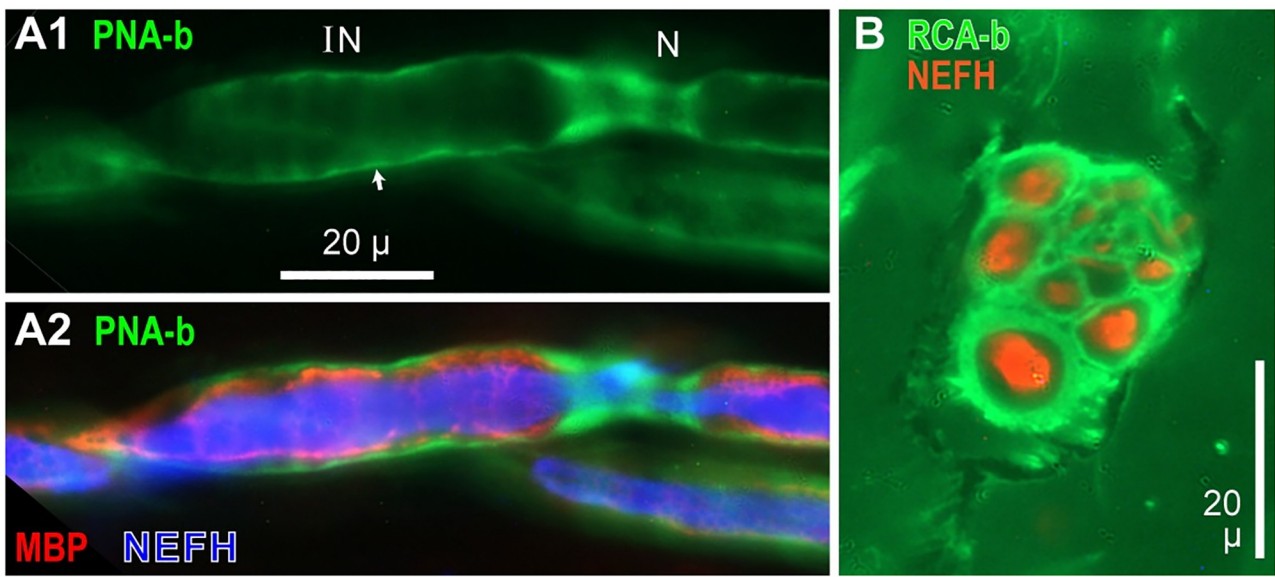

**Fig 11. Lectin labeling of axon sheaths.** (**A1**) Lectin PNA-biotin (PNA-b) labeled a thin sheath (arrow) around large myelinated ALLn afferent axons, surrounding internodes (IN) and continuing across nodes (N); 40x lens. (**A2**) Superimposed triple label of same axons. Myelin (red) was labeled by anti-myelin basic protein (MBP), and electroreceptor afferent axons (blue) by anti-neurofilament-H (NEFH). (**B**) Lectin RCA120-biotin (green) labeled perineurium tissue of a small nerve. Orange: Individual axons labeled by anti-neurofilament-H (NEFH).

Lectin RCA$_{120}$-biotin labeled the interstitial perineurium tissue of small nerve trunks near AOs (Fig 11B).

## Lectin label of rostrum skin

We imaged lectin label of the specialized skin on *Polyodon*'s rostrum, ~1 mm thick, which contributes to electrosensitivity (like the skin of other freshwater fish [1]). This headshield skin included a striated ectoderm epidermis, loose dermis connective tissue between AOs laterally, and deeper hypodermis over subsurface cartilage plates.

Lectin WGA labeled throughout the striated ectoderm (SE, Figs 1, 7 and 10), including several superficial squamous cell layers, separated cells in middle layers, and a basal layer of cuboidal germinal cells delimited by a ruffled basal lamina (joined to dermis). The exterior surface of striated ectoderm was maximally WGA$^+$. Lectin SBA labeled similarly but less. Both lectins brightly labeled goblet organs in striated ectoderm, containing large vesicles of skin mucus.

Biotinylated lectins LEL, PNA, or UEA1 left largely unstained the striated ectoderm (except goblet organs).

## Discussion

We surveyed the spatial binding of 8 lectins within ampullary organs of electroreceptors on *Polyodon*'s rostrum. Lectins continue to be useful probes for staining organs, tissues, cells, glycocalyces, and secretions [17, 18, 24–28, 34–38]. Lectins have been used to label other sensory receptors of fish including the taste, olfactory, and vomeronasal epithelia [39–42]. While lectins have recognition domains for specific carbohydrates, attached to macromolecules or cell surfaces, lectin binding may also depend on a carbohydrate molecule's stereo geometry, position, linkage types and branching, or adjacent chemical groups [17, 18, 43, 44]. Also, a lectin may bind to alternate ligands. For example, WGA binds to N-acetyl-D-glucosamine and its ß-

linked dimers and trimers [17, 18], invoked here, but WGA also binds to groups of terminal sialic acids, e.g., of neuraminic acids [45].

We used different conjugates of lectins. Biotinylated lectins were effective at low concentrations (1–5 μg/L). Some lectin-fluor conjugates required ~10x higher concentrations.

'Hard' tissue fixation may impair lectin access and binding, but was required to image upright microvilli on EN support cells apically. Brief fixation yielded superior lectin label within AO cells.

Proteomic analysis of proteoglycans in elasmobranch AoL gel [7] reported peptide fragments matching mucins, serotransferrin, keratin, actin, tropomyosin, and numerous other proteins. Several proteins, including mucins, are core proteins carrying numerous keratan sulfate sidechains [7] containing abundant N-acetyl-D-glucosamine (to which WGA could bind). Keratan sulfates endow AoL gel with high conductivity [3, 7, 46]. Carbohydrates form globular aggregates in the hydrogel [47]. We attempted to label *Polyodon* AO gel with two different polyclonal antibodies to human mucins (Abcam, genes MUC1, MUC2), with negative results (perhaps explained by species differences). An anonymous reviewer contributed comments about mucins (see website).

## Lectin label of AO gel and EN support cells

We labeled the gel in *Polyodon* AOs with lectins WGA or SBA, as either -biotin or -fluor conjugates (Figs 1, 2, 4C and 7B). Their binding to gel and AOs was abolished by preabsorption with respective amino-carbohydrate ligands (Fig 2). WGA or SBA labeled EN support cells, including their apical microvilli (Fig 4), cytoplasmic vesicles and organelles (Fig 4), and their apical surfaces (Figs 3 and 7I). Our imaging provided evidence that EN support cells, especially their apical microvilli, are major secretors of AO gel components carrying N-acetyl-D-glucosamine or N-acetyl-D-galactosamine residues (ligands of WGA or SBA respectively). These amino-carbohydrates were also identified in the canal gel of AoL of marine sharks and rays [4–7], consistent with homology of *Polyodon* AOs [8–11]. Our labeling data for SBA were conclusive, but we had two concerns for the basis of WGA labeling.

i. In our preabsorption controls using chitin hydrolysate solution, the final concentration of NaCl was ~2.5 M (estimated) in mixtures with WGA-biotin or other lectins, applied to tissue sections. High ionic strength is typically a stringent condition for molecular binding. However, binding of WGA to chitosans is insensitive to high ionic strength, e.g., in 1.5 M NaCl [48]. Also, the positive label of AOs by other biotinylated lectins was little-altered by similar preabsorption mixtures with chitin hydrolysate solution. For example, label by preabsorbed SBA-b (Fig 1B$_1$) appeared unaltered compared to label with SBA-b in buffer (Fig 2E); both labeled AO gel.

ii. As noted, WGA may bind to alternate ligands. Glucosamine, the ligand invoked here, comprised mean ~65% of total carbohydrate in AoL canal gel of 16 elasmobranch species [5], and N-acetyl-D-glucosamine is a chief component of abundant keratan sulfate sidechains of proteoglycans in AoL gel [7]. However, another ligand of WGA, clustered terminal sialic acids, was also noted in AoL gel [7]. Further study is needed to distinguish these possible alternates as ligands for WGA binding at sites in *Polyodon* AOs.

Visible striations of AO gel leading to a *Polyodon* EN (Fig 4 in Jørgensen et al. [14]) were consistent with gel secretion from the EN.

We estimated lectin label of EN cells based on apparent fluorescence brightness in a lectin's color channel. The hourglass or mushroom-like shape of support cells meant that most cytoplasm was apical, so enhanced brightness may be expected apically in lectin+ support cells. A special case was the most apical cytoplasm of support cells, which covered a similar EN area as

their microvilli. Hence the brighter $WGA^+/SBA^+$ label of microvilli (Fig 4A$_2$ and inset, Fig 4D) was not due to the shape of support cells.

Our imaging and attempts at morphometry were inconclusive as to whether other lectins may bind to AO gel. Candidates included PNA which labeled apical microvilli on support cells (Fig 5A), and RCA$_{120}$ which labeled apically in EN (Fig 7G). The ligand of UEA1 (L-fucose) is a minor component of AoL gel [6, 7].

The flattened epithelial cells of ampulla wall appeared mainly structural or defensive. Additional secretions may come from distal wall cells and neck cells, which had projecting barbs or papillas that may secrete defensive molecules (below).

Concentrated luminal gel was clearly preserved at deeper levels of AOs (Figs 1, 2 and 7B). Concentrated gel likely continues to an AO's pore, and beyond. Fig 1B$_1$ showed concentrated gel near a pore, albeit torn. Outside of pores, expelled gel (including bright particles: *, Fig 7B) was visible in some sectioned AOs. Profile plots (Fig 1) of lectin-labeled gel in AOs tended to show declining gel fluorescence, from the EN to the pore. Explanations may include: (i) loss or disturbance of gel during tissue processing, (ii) dilution of AO gel by external water, or (iii) autolysis of gel.

Colloidal biological gels are fragile and difficult to preserve. Here, possible artifacts included: (i) sectioning damage, (ii) shrinkage of gel (orange ■, Fig 1A$_1$ and 1A$_2$), or (iii) loss of AO gel or tissue during labeling steps. We immersed specimen blocks in fixative (METHODS), in an attempt to preserve gel material near AO pores. Sectioned gel was supported by adhesion to slides. Nonetheless, AO gel may have been lost, or only part remained (Fig 7A) [12].

## Gel clearance model

Different sites of gel secretion can be envisioned in ampullary ERs. In marine AoL, gel is secreted from canal wall epithelia, and from the ampulla [2, 13, 49]. Gel from the ampulla (Cap, Fig 12A) would set an initial flow rate of gel, e.g., on a scale of $\mu^3$/h. To this would add gel from canal wall [2]. Canals can be quite long, e.g., 1 m in a large ray [2]. If gel were secreted uniformly along a canal of fixed diameter, then the flow (volume/time) of gel passing a cross-section of canal would increase with distance, going distally from the ampulla (Fig 12A), before exit at the skin pore. This implies a distal increase of gel flow velocity, e.g., on a scale of $\mu$/h, in canals of fixed width. Most images of AoL show straight tubular canals. However, some canals increase in diameter distally [2, 50]. Such distension may compensate to hold velocity ~constant as gel flows distally in elasmobranch AoL canals.

In *Polyodon* AOs, which lack canals, we suggest a variant model (Fig 12B): the gel filling AOs is secreted mainly by support cells of the large EN at the basal interior pole of an AO, then gel is pushed towards the skin pore (distally), and shed. Similar gel flow likely occurs in related sturgeon AOs [12]. This 'gel clearance' model lends an advantage of continuously renewing 100% of the gel in an AO, and pushing 'old' gel out the skin pore. The basal EN placement, and its maximal width, guarantee that the entire interior volume of an AO is swept by gel flow. This likely defends the AO lumen from entry of microbes or small parasites (below).

From this model, the flow (volume / time, $\mu^3$/h) of gel would be constant distal of an EN in *Polyodon* AOs, passing any given cross-sectional level of ampulla wall (flat red line, Fig 12B). Also, the AO lumen presumably would have a positive pressure (turgor), relative to exterior water, due to gel secretion and impedance of gel flow at the narrowed neck.

## Gel velocity model

Acceleration of gel flow likely occurs distally in a *Polyodon* AO, due to geometric taper of the ampulla wall as it narrows to a neck and pore. Within the lumen of an AO, gel was assumed to

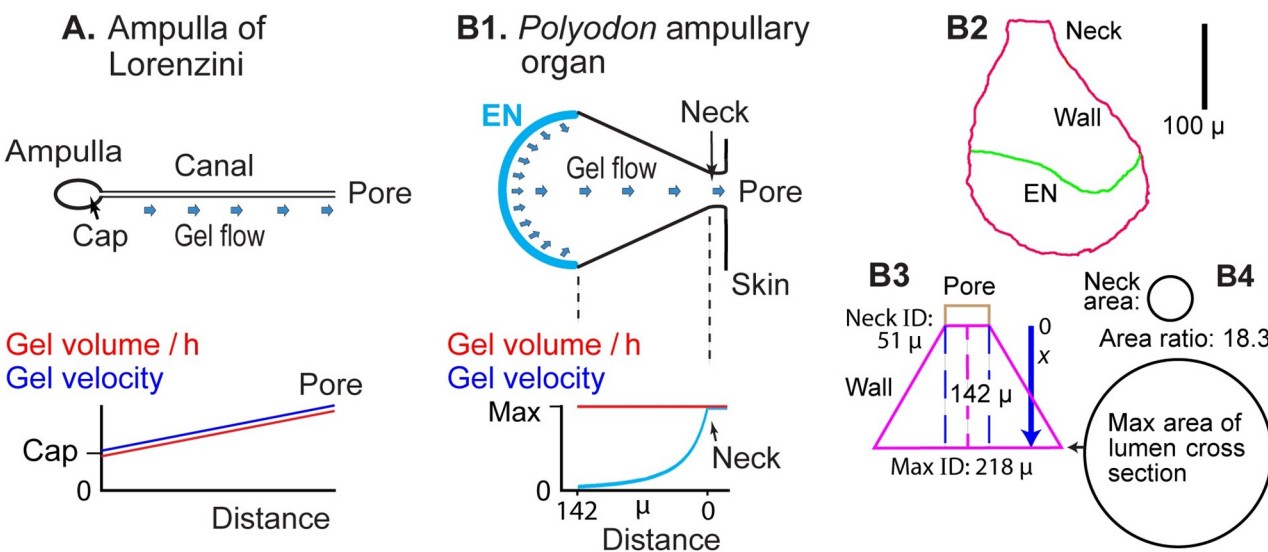

**Fig 12. Models of gel secretion in ampullary electroreceptors.** (**A**) Ampulla of Lorenzini: Blue arrows indicates gel secretion from canal wall cells, along the length of a canal. Cap: The ampulla's centrum cap, near the start of a canal, also secretes gel [2, 13]. (**B1–4**) Model of a small AO of *Polyodon*, based on Fig 10A. (**B1**) Arrows indicate gel secretion mainly from support cells of the wide EN covering the interior basal pole of each AO. Gel accelerates going toward the pore, due to AO taper. Relative gel velocity (blue curve) was calculated as inversely proportional to the circular cross-sectional area of the ampulla lumen (Tab 1 in S2 Table). (**B2**) Traced outline of the small AO in Fig 10A. (**B3**) Simplified scale model of the ampulla wall (pink lines) as a symmetrical funnel-shaped taper, based on the pink dimensional lines in Fig 10A. (**B4**) To-scale areas of the neck and the maximal ampulla wall width, assumed circular.

flow from an EN towards the neck (that is, distally) as an incompressible fluid, with local velocities on a scale of μ/h. Minimum gel velocity $V_{min}$ was expected near an EN, arising from ongoing gel secretion from support cells; the EN width often corresponded to the maximum width ($W_{max}$) of the AO lumen in our cross-section images. Maximum gel velocity $V_{max}$ was expected at the neck, due to its narrowed luminal width ($W_N$, 40–150 μ).

While we did not measure the actual velocity of AO gel, its relative velocity could be calculated, normalized to $V_{max}$ at the narrowed neck. We modeled relative gel velocity $V_x/V_{max}$ over the taper of the ampulla wall (Tab 1 in S2 Table) as varying inversely with the ratio of the AO lumen's cross-section area $A_x$ to the neck's minimal area $A_N$: $V_x/V_{max}$ 1($A_x/A_N$). Relative $V_x$ was calculated starting from the neck: $V_x/V_{max} = 1$ @ x = 0. The total increase of relative gel velocity, from EN to neck, equaled the areal ratio $A_{max}/A_N$ (Fig 12B4).

A nonlinear increase of relative gel velocity shown in Fig 12B1 (blue curve) was based on the small AO in Fig 10A (pink measurement lines), traced in Fig 12B2, and geometrical calculations (Tab 1 in S2 Table). The shape of the ampulla wall was simplified to a symmetrical linear taper (Fig 12B3). The model reported that gel velocity would increase ~18-fold in this example, due to change of luminal cross section area, from the EN to the neck (Fig 12B4). For a linear wall taper, gel accelerates continuously (distal of the EN), most notably near the neck. The profile of velocity increase would depend on the AO wall shape; Fig 12B considers only a case of symmetrical linear wall taper.

Similarly, for a sample of 30 *Polyodon* AOs (Tab 2 in S2 Table), the mean relative increase in gel velocity due to wall taper, $V_{max}/V_{min} = A_{max}/A_N$, was estimated as 11 ± 8.7 (range 2.7– 42.8). For this, we measured luminal widths $W_{max}$ and $W_N$ (as in the photo in Tab 2 in S2 Table) in cross-section images of the 30 AOs, each showing an open pore and no subdivisions. An open pore implied that a section passed near the middle of an AO, where AO dimensions were maximal. Even so, the reported neck widths were likely too small, due to bias of section

sampling (off-center sections were more probable), tending to overestimate $A_{max}/A_N$. The tilted EN orientation in many AOs (Fig 7C) would increase the maximum AO cross section, and increase geometric acceleration. Our sample of AOs included distinct size classes, including small rounded AOs which may develop into deep elongated AOs (Fig 7D).

This model was scalar. The direction of gel flow and velocity was generally distal: from a basal EN, along the long axis of an AO, into the neck and pore distally, and out. Estimating a vector field of gel flow was beyond the scope here. Our model did not consider wall drag or impedance to gel flow.

Most AOs on the rostrum did taper to a narrowed pore, in our cross-section images, but some AOs tapered little. We concluded that geometric acceleration of gel occurs in most AOs of *Polyodon*, and presumably in sturgeon AOs also. In elasmobranch AoL whose canals are distended distally (above), similar geometric acceleration of internal gel may occur at narrowed skin pores.

## Defenses of ampullary organs

The >50,000 AOs of a paddlefish [15] present significant risk of infection, as each AO's skin pore provides an open portal for possible entry of pathogens / parasites into AOs or the body interior. Hence defense against invasion by microbes or small parasites is vital for AOs. Besides gel clearance, defenses may include anti-pathogen molecules in AO gel, e.g., serotransferrin [7]. The numerous small papillas lining the neck and pore (P, Fig 10) may secrete or be coated with defensive molecules, as may the barbs on transitional epithelial cells of ampulla wall (B, Fig 7B–7D). The functions of papillas and barbs in *Polyodon* AOs are unknown, but their large numbers and strategic location at the entrance to each AO indicate a defensive role [31]. The distinct sequential bands of papillas vs. barbs may combat different threats.

Diseases and parasites of paddlefish were reviewed by Durborow, Kuchta & Scholz [51]. Several species of large nematodes have been found in the buccal and alimentary cavities of paddlefish [52, 53]. In other fish, small copepod parasites may invade lateral line canals [54–56]. In general, ducted cutaneous organs are at risk of microbe / parasite invasion.

## Supporting information

**S1 Fig. Autofluorescence of EN cells.**
(DOCX)

**S1 Table. Measured values, data calculations.**
(XLSX)

**S2 Table. Geometric model of ampulla wall taper and relative gel velocity.**
(XLSX)

## Author Contributions

**Conceptualization:** David F. Russell, Wenjuan Zhang.

**Data curation:** David F. Russell, Lilia L. Neiman.

**Formal analysis:** David F. Russell, Thomas C. Warnock.

**Funding acquisition:** David F. Russell.

**Investigation:** David F. Russell, Wenjuan Zhang, Lilia L. Neiman.

**Methodology:** David F. Russell, Lilia L. Neiman.

**Project administration:** David F. Russell.

**Resources:** David F. Russell.

**Software:** David F. Russell.

**Supervision:** David F. Russell.

**Validation:** David F. Russell, Thomas C. Warnock, Lilia L. Neiman.

**Visualization:** David F. Russell, Lilia L. Neiman.

**Writing – original draft:** David F. Russell.

**Writing – review & editing:** David F. Russell.

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
