## [Decision Letter · Decision Letter 0]

28 Feb 2022

PONE-D-22-01494Lectin binding and gel secretion within Lorenzinian electroreceptors of *Polyodon**PLOS ONE*

*Dear Dr. Russell,*

*Thank you for submitting your manuscript to PLOS ONE. After careful consideration, we feel that it has merit but does not fully meet PLOS ONE’s publication criteria as it currently stands. Therefore, we invite you to submit a revised version of the manuscript that addresses the points raised during the review process.*

*Please submit your revised manuscript by Apr 14 2022 11:59PM. If you will need more time than this to complete your revisions, please reply to this message or contact the journal office at plosone@plos.org. *

*Please include the following items when submitting your revised manuscript:*

*A rebuttal letter that responds to each point raised by the academic editor and reviewer(s). You should upload this letter as a separate file labeled 'Response to Reviewers'.*

*A marked-up copy of your manuscript that highlights changes made to the original version. You should upload this as a separate file labeled 'Revised Manuscript with Track Changes'.*

*An unmarked version of your revised paper without tracked changes. You should upload this as a separate file labeled 'Manuscript'.*

**

*If applicable, we recommend that you deposit your laboratory protocols in protocols.io to enhance the reproducibility of your results. Protocols.io assigns your protocol its own identifier (DOI) so that it can be cited independently in the future. For instructions see: https://journals.plos.org/plosone/s/submission-guidelines#loc-laboratory-protocols. Additionally, PLOS ONE offers an option for publishing peer-reviewed Lab Protocol articles, which describe protocols hosted on protocols.io. Read more information on sharing protocols at https://plos.org/protocols?utm_medium=editorial-email&utm_source=authorletters&utm_campaign=protocols.*

*We look forward to receiving your revised manuscript.*

*Kind regards,*

*Thomas Abraham, PhD*

*Academic Editor*

*PLOS ONE*

*Journal Requirements:*

2. To comply with PLOS ONE submissions requirements, in your Methods section, please provide additional information on the animal research and ensure you have included details on (1) methods of sacrifice, (2) methods of anesthesia and/or analgesia, and (3) efforts to alleviate suffering.

"Supported by NIH grant 5R21GM103494 to DFR, by research funds from Ohio University, and by a Provost Undergraduate Research Fund grant to WZ."

We note that you have provided funding information. However, funding information should not appear in the Funding section or other areas of your manuscript. We will only publish funding information present in the Funding Statement section of the online submission form. 

"Supported by NIH grant 5R21GM103494 to DFR, by research funds from Ohio University, and by a Provost Undergraduate Research Fund grant to WZ. The funders had no role in study design, data collection and analysis, decision to publish, or preparation of the manuscript."

*5. Please review your reference list to ensure that it is complete and correct. If you have cited papers that have been retracted, please include the rationale for doing so in the manuscript text, or remove these references and replace them with relevant current references. Any changes to the reference list should be mentioned in the rebuttal letter that accompanies your revised manuscript. If you need to cite a retracted article, indicate the article’s retracted status in the References list and also include a citation and full reference for the retraction notice.*

**

*Reviewers' comments:*

*Reviewer's Responses to Questions*

*

**Comments to the Author**
*

*1. Is the manuscript technically sound, and do the data support the conclusions?*

*The manuscript must describe a technically sound piece of scientific research with data that supports the conclusions. Experiments must have been conducted rigorously, with appropriate controls, replication, and sample sizes. The conclusions must be drawn appropriately based on the data presented. *

*Reviewer #1: Yes*

*2. Has the statistical analysis been performed appropriately and rigorously? *

*Reviewer #1: Yes*

*3. Have the authors made all data underlying the findings in their manuscript fully available?*

*The PLOS Data policy requires authors to make all data underlying the findings described in their manuscript fully available without restriction, with rare exception (please refer to the Data Availability Statement in the manuscript PDF file). The data should be provided as part of the manuscript or its supporting information, or deposited to a public repository. For example, in addition to summary statistics, the data points behind means, medians and variance measures should be available. If there are restrictions on publicly sharing data—e.g. participant privacy or use of data from a third party—those must be specified.*

*Reviewer #1: Yes*

*4. Is the manuscript presented in an intelligible fashion and written in standard English?*

*PLOS ONE does not copyedit accepted manuscripts, so the language in submitted articles must be clear, correct, and unambiguous. Any typographical or grammatical errors should be corrected at revision, so please note any specific errors here.*

*Reviewer #1: Yes*

*5. Review Comments to the Author*

*Please use the space provided to explain your answers to the questions above. You may also include additional comments for the author, including concerns about dual publication, research ethics, or publication ethics. (Please upload your review as an attachment if it exceeds 20,000 characters)*

*Reviewer #1: 1. There is no description of the preabsorption method in Materials and Methods section. Is the lectin absorbed from the reaction solution by the resin on which the sugar is immobilized and then used for staining? Or did the authors perform lectin-staining in the presence of haptenic sugar? If the latter case, I think inhibition in the presence of haptenic sugar is better than preabsorption. Furthermore, the conditions for inhibition by the above haptenic sugar are not described. Please describe what concentration ("in excess" in Results, line 272) the sugar was added.*

*2. Chitin is the main component of insects and bacteria covering the outside of body or cells, respectively, but it does not exist in fish. WGA actually binds strongly to chitin, but WGA ligands in fish cells or tissues thought to be clustered sialic acids attached on O-linked glycans of glycoproteins such as mucins (1). Therefore, the observation that AO gel was stained with WGA or SBA is considered to be a mucin-like secreted proteins. It would be better to add such information to Discussion section.*

*(1) Bhavanandan VP and Katlic AW. J Biol Chem 254, 4000-4008 (1979)*

*3. lines 276 and 279*

*"N-acetyl-D-galactose" should be changed to "N-acetyl-D-galactosamine".*

*4. The following information about lectin-binding specificity may be helpful for the readers of this paper. If the authors also think so, add the information to Discussion section.*

*RCA120 binds to beta galactose residues present at the non-reducing terminal of both N-linked and O-linked oligosaccharides, but the binding is attenuated by the further addition of sialic acids at beta galactose residues (2).*

*(2) Baenziger JU and Fiete D. J Biol Chem 254, 9795-9799 (1979)*

*Since Con A binds only to N-linked glycans (3) on the surface of the cell, the staining data using Con A is thought to stain common glycoproteins present on the cell surface.*

*(3) Kurusius T et al. FEBS Lett. 71, 117-120 (1976)*

*Mucin-like glycoproteins are often secreted from secretory tissues and have many Ser/Thr residues attached with O-linked glycans. By contrast, Mucin-like glycoproteins do not have so many N-glycosylation sites, and they generally do not have much N-linked glycans to which Con A could bind.*

*6. PLOS authors have the option to publish the peer review history of their article (what does this mean?). If published, this will include your full peer review and any attached files.*

**

**

*Reviewer #1: No*

**

*While revising your submission, please upload your figure files to the Preflight Analysis and Conversion Engine (PACE) digital diagnostic tool, https://pacev2.apexcovantage.com/. PACE helps ensure that figures meet PLOS requirements. To use PACE, you must first register as a user. Registration is free. Then, login and navigate to the UPLOAD tab, where you will find detailed instructions on how to use the tool. If you encounter any issues or have any questions when using PACE, please email PLOS at figures@plos.org. Please note that Supporting Information files do not need this step.*

---

## [Author Response · Author response to Decision Letter 0]

27 Aug 2022

Response to reviewer

1. Preabsorption method: This is now described in detail in the Results text section “Lectin controls”. Also, we moved preabsorption imaging data into Fig 2 instead of ex-supplement 1. A lectin solution was pre-mixed with inhibiting sugar and incubated for 1-2 hours, and then the mixture was applied to a tissue slide so that lectin binding occurred in the presence of inhibiting sugar, as recommended by the reviewer. 

The reviewer required stating the concentrations of preabsorption sugars. We now supply this value for N acetyl-D-galactosamine (100 mM). We also used a complex sugar mixture, chitin hydrolysate solution (Vector Labs SP 0090), in other preabsorption controls. However, Vector Labs’ data sheet did not state the concentrations of its several component sugars, including haptenic N acetyl-D-glucosamine and various oligomers of it. That is, chitin hydrolysate was a complex degradation mixture from chitin, incompletely characterized but widely used with lectins. We mixed lectins with a low dilution of chitin hydrolysate solution. Therefore, we can only state that the component sugars of chitin hydrolysate were “in excess”.

2. “Chitin is the main component of insects and bacteria covering the outside of body or cells, respectively, but it does

not exist in fish.” 

The reviewer keyed on the term ‘chitin’, an arthropod polymer. However, what we used for preabsorption of WGA was chitin hydrolysate solution, consisting of haptenic N-acetyl-D-glucosamine and oligomers of this amino sugar, to which WGA can bind. Chitin was merely the source of the hydrolysate. To clarify this, we substituted “chitin hydrolysate” for “chitin” throughout the text, and also in panel labels of Fig 2. 

“WGA actually binds strongly to chitin [N-acetyl-D-glucosamine and oligomers], but WGA ligands in fish cells or tissues thought to be clustered sialic acids attached on O-linked glycans of glycoproteins such as mucins”

The reviewer pointed out that WGA also binds to terminal sialic acid groups, and supplied the original reference for this, now added to the article. We previously cited this issue in Discussion, but now mention it in the Results (page 12) and address it explicitly in paragraphs #1 and #8 of the Discussion. We point out in paragraph #8 that glucosamine is highly concentrated (65% of total carbohydrate) in the gel filling homologous ampullary organs. 

“Therefore, the observation that AO gel was stained with WGA or SBA is considered to be a mucin-like secreted proteins.”

We added paragraph #5 to the Discussion citing recent structural analyses [ref 7] on glycoproteins in the gel filling Lorenzinian ampullary organs, which found peptide fragments of mucin-like proteins. However, this need not imply WGA binding to sialic acids on mucin, as the reviewer asserted. Instead, ref 7 indicated sparse sialic acids, but found abundant N-acetyl-D-glucosamine in numerous keratan sulfate sidechains on mucin and other proteins, consistent with N-acetyl-D-glucosamine and its oligomers as a basis for our observed label of Polyodon’s ampullary organ gel by WGA. 

3. “N acetyl-D-galactosamine” and “N acetyl-D-glucosamine” are now used uniformly throughout the article. 

4. We thank the reviewer for original references, now added to the Discussion, about ligands of lectins RCA and ConA. Our previous Results and Discussion text attempted to compare labeling by these two lectins, but now this has been removed and the related text reorganized, based on our recent lab work. 

Following the reviewer’s suggestion, we included part of their commentary as Discussion paragraph #6, in quotation marks. 

From the Cover Letter for revised manuscript

Journal Requirements:

1. The MS was reformatted to match PLOS1’s styles. 

2. Additional information about animal procedures has been included in the Methods section to address 

(1) methods of sacrifice, and (2) methods of anesthesia. (3) As noted, the animals were under deep anesthesia, preventing suffering. 

3. Mention of funding was removed from the text. The funding acknowledgement has been changed:

" Supported by NIH grant 5R21GM103494 to DFR, by research funds from Ohio University, and by a Provost Undergraduate Research Fund grant to WZ. We thank Jeffrey B. Thuma for collecting confocal images. The funders had no role in study design, data collection and analysis, decision to publish, or preparation of the manuscript. " 

4. Titles for Supplementary information are listed at the end of the manuscript. Supplements 1, 2, 3 were updated from ex-supplements 2, 4, 5 respectively. The in-text citations of supplements and figures were updated. 

5. References and DOI links have been rechecked. The number of references was increased from 40 to 54. 

6. Table 1 and figure legends were embedded in the text.

---

## [Decision Letter · Decision Letter 1]

17 Oct 2022

Lectin binding and gel secretion within Lorenzinian electroreceptors of *Polyodon*

*PONE-D-22-01494R1*

*Dear Dr. Russell,*

*We’re pleased to inform you that your manuscript has been judged scientifically suitable for publication and will be formally accepted for publication once it meets all outstanding technical requirements.*

*Within one week, you’ll receive an e-mail detailing the required amendments. When these have been addressed, you’ll receive a formal acceptance letter and your manuscript will be scheduled for publication.*

*An invoice for payment will follow shortly after the formal acceptance. To ensure an efficient process, please log into Editorial Manager at http://www.editorialmanager.com/pone/, click the 'Update My Information' link at the top of the page, and double check that your user information is up-to-date. If you have any billing related questions, please contact our Author Billing department directly at authorbilling@plos.org.*

*If your institution or institutions have a press office, please notify them about your upcoming paper to help maximize its impact. If they’ll be preparing press materials, please inform our press team as soon as possible -- no later than 48 hours after receiving the formal acceptance. Your manuscript will remain under strict press embargo until 2 pm Eastern Time on the date of publication. For more information, please contact onepress@plos.org.*

*Kind regards,*

*Hideshi Okada, M.D., Ph.D.*

*Academic Editor*

*PLOS ONE*

* *

*Additional Editor Comments (optional):*

* *

*Reviewers' comments:*

*Reviewer's Responses to Questions*

*

**Comments to the Author**
*

*1. If the authors have adequately addressed your comments raised in a previous round of review and you feel that this manuscript is now acceptable for publication, you may indicate that here to bypass the “Comments to the Author” section, enter your conflict of interest statement in the “Confidential to Editor” section, and submit your "Accept" recommendation.*

*Reviewer #1: All comments have been addressed*

*Reviewer #2: All comments have been addressed*

*2. Is the manuscript technically sound, and do the data support the conclusions?*

*The manuscript must describe a technically sound piece of scientific research with data that supports the conclusions. Experiments must have been conducted rigorously, with appropriate controls, replication, and sample sizes. The conclusions must be drawn appropriately based on the data presented. *

*Reviewer #1: Yes*

*Reviewer #2: Yes*

*3. Has the statistical analysis been performed appropriately and rigorously? *

*Reviewer #1: Yes*

*Reviewer #2: Yes*

*4. Have the authors made all data underlying the findings in their manuscript fully available?*

*The PLOS Data policy requires authors to make all data underlying the findings described in their manuscript fully available without restriction, with rare exception (please refer to the Data Availability Statement in the manuscript PDF file). The data should be provided as part of the manuscript or its supporting information, or deposited to a public repository. For example, in addition to summary statistics, the data points behind means, medians and variance measures should be available. If there are restrictions on publicly sharing data—e.g. participant privacy or use of data from a third party—those must be specified.*

*Reviewer #1: Yes*

*Reviewer #2: Yes*

*5. Is the manuscript presented in an intelligible fashion and written in standard English?*

*PLOS ONE does not copyedit accepted manuscripts, so the language in submitted articles must be clear, correct, and unambiguous. Any typographical or grammatical errors should be corrected at revision, so please note any specific errors here.*

*Reviewer #1: Yes*

*Reviewer #2: Yes*

*6. Review Comments to the Author*

*Please use the space provided to explain your answers to the questions above. You may also include additional comments for the author, including concerns about dual publication, research ethics, or publication ethics. (Please upload your review as an attachment if it exceeds 20,000 characters)*

*Reviewer #1: I have read the revised version of the manuscript. I made several comments about lectins last time, and these comments were properly addressed and reflected in the revised manuscript. Therefore, I think that this version of the manuscript becomes acceptable for publification on PLoS One.*

*Reviewer #2: The authors have addressed all the comments. The quality of the manuscript has been improved. Thus, I recommend the acceptance of this manuscript.*

*7. PLOS authors have the option to publish the peer review history of their article (what does this mean?). If published, this will include your full peer review and any attached files.*

**

**

*Reviewer #1: No*

*Reviewer #2: No*

---

## [Editor Report · Acceptance letter]

8 Nov 2022

PONE-D-22-01494R1 

Lectin binding and gel secretion within Lorenzinian electroreceptors of *Polyodon*

Dear Dr. Russell:

I'm pleased to inform you that your manuscript has been deemed suitable for publication in PLOS ONE. Congratulations! Your manuscript is now with our production department. 

Kind regards, 

on behalf of

Dr. Hideshi Okada 

Academic Editor

PLOS ONE